# Orexin receptors 1 and 2 in serotonergic neurons differentially regulate peripheral glucose metabolism in obesity

Xing Xiao [1,2,3], Gagik Yeghiazaryan[3,4], Simon Hess[3,4], Paul Klemm [1,2,3], Anna Sieben [1,2,3], André Kleinridders[1,2,3,5,10], Donald A. Morgan[6], F. Thomas Wunderlich [1,2,3], Kamal Rahmouni[6], Dong Kong [7,8], Thomas E. Scammell[9], Bradford B. Lowell[8], Peter Kloppenburg[3,4], Jens C. Brüning [1,2,3,5] & A. Christine Hausen [1,2,3 ✉]

The wake-active orexin system plays a central role in the dynamic regulation of glucose homeostasis. Here we show orexin receptor type 1 and 2 are predominantly expressed in dorsal raphe nucleus-dorsal and -ventral, respectively. Serotonergic neurons in ventral median raphe nucleus and raphe pallidus selectively express orexin receptor type 1. Inactivation of orexin receptor type 1 in serotonin transporter-expressing cells of mice reduced insulin sensitivity in diet-induced obesity, mainly by decreasing glucose utilization in brown adipose tissue and skeletal muscle. Selective inactivation of orexin receptor type 2 improved glucose tolerance and insulin sensitivity in obese mice, mainly through a decrease in hepatic gluconeogenesis. Optogenetic activation of orexin neurons in lateral hypothalamus or orexinergic fibers innervating raphe pallidus impaired or improved glucose tolerance, respectively. Collectively, the present study assigns orexin signaling in serotonergic neurons critical, yet differential orexin receptor type 1- and 2-dependent functions in the regulation of systemic glucose homeostasis.

[1] Max Planck Institute for Metabolism Research, Department of Neuronal Control of Metabolism, Cologne, Germany. [2] Center for Endocrinology, Diabetes and Preventive Medicine (CEDP), University Hospital Cologne, Cologne, Germany. [3] Excellence Cluster on Cellular Stress Responses in Aging Associated Diseases (CECAD) and Center for Molecular Medicine Cologne (CMMC), University of Cologne, Cologne, Germany. [4] Biocenter, Institute for Zoology, University of Cologne, Cologne, Germany. [5] National Center for Diabetes Research (DZD), Neuherberg, Germany. [6] Department of Neuroscience and Pharmacology, University of Iowa, Carver College of Medicine, Iowa City, IA, USA. [7] Division of Endocrinology, Department of Pediatrics, F.M. Kirby Neurobiology Center, Boston Children's Hospital and Harvard Medical School, Boston, MA, USA. [8] Division of Endocrinology, Diabetes and Metabolism, Department of Medicine, Beth Israel Deaconess Medical Center and Harvard Medical School, Boston, MA, USA. [9] Department of Neurology, Beth Israel Deaconess Medical Center and Harvard Medical School, Boston, MA, USA. [10]Present address: Institute of Nutritional Science, Department of Molecular and Experimental Nutritional Medicine, University of Potsdam, Nuthetal, Germany. ✉email: christine.hausen@sf.mpg.de

The obesity epidemic represents a major global socio-economic burden, currently affecting over a third of the world's population[1]. However, the underlying cellular causes of weight gain and its associated metabolic comorbidities, such as type 2 diabetes mellitus, are still only poorly understood[2]. Orexins (hypocretins) are neuropeptides expressed by distinct neurons in the lateral hypothalamic area (LHA) and adjacent regions, including the perifornical area, and dorsomedial and posterior hypothalamus in humans and rodents[3]. The orexin system consists of the two distinct neuropeptides orexin-A and orexin-B (hypocretin-1 and -2), which are both derived from a common precursor peptide, and they act on two G-protein coupled receptors, orexin receptor type 1 (Ox1R) and type 2 (Ox2R)[4,5]. Human cerebrospinal fluid (CSF) orexin negatively correlates with serum glucose levels, and narcolepsy patients exhibit low orexin levels in CSF and increased obesity[6,7]. Ablation of orexin neurons in mice results in narcolepsy, hypoactivity, and late-onset obesity with or without hypophagia[8,9]. Mice that ubiquitously overexpress orexin show an increase in energy expenditure and are resistant to diet-induced obesity[10]. Interestingly, pharmacogenetic stimulation of orexin neurons results in an elevation of blood glucose levels with a simultaneous increase in locomotor activity and food intake[11]. The wake-active orexin system[12] plays a key role in maintaining dynamic regulation and daily rhythm of peripheral glucose homeostasis, and it has been described to have dual roles in the regulation of blood glucose, possibly through the autonomic nervous system[13,14].

Differential regulatory roles of Ox1R and Ox2R in energy homeostasis and glucose metabolism throughout the organism have been revealed employing pharmacological or genetic methods. One study shows that Ox2R but not Ox1R predominantly mediates the orexin-overexpression-induced amelioration of obesity in mice fed a high-fat diet (HFD)[10]. In another study, Ox1R deficiency increases the body weight of mice fed a chow diet, but decreases intake of HFD and prevents HFD-induced excess of body weight gain, while Ox2R deficient mice mainly exhibited decreased energy expenditure[15]. Moreover, systemic histological studies have shown overlapping but distinct expressions of Ox1R and Ox2R in many brain regions, such as the cerebral neocortex, basal ganglia, hippocampus, thalamus, midbrain, and reticular formation[16–18]. Fasting increases Ox1R mRNA in the ventral medial hypothalamic nucleus, while Ox2R mRNA is increased in the arcuate nucleus of the hypothalamus[19].

Interestingly, within a defined cell population of serotonergic neurons in raphe nuclei (RN), RNA sequencing (RNA-seq) experiments revealed that mainly Ox2R but not Ox1R is differentially expressed among the subpopulations in dorsal raphe nucleus (DR), median raphe nucleus (MR) and caudal raphe nuclei including the raphe pallidus nucleus (RPa)[20]. Serotonergic neurons in subregions of RN have been described as very heterogeneous in aspects of morphology, electrophysiological behavior, projections, and functions[20,21].

Importantly, the serotonergic system plays a central role in regulating wakefulness, appetite, glucose metabolism, and energy homeostasis[22–25]. Pharmacological experiments show that serotonin inhibits food intake and clinical drugs targeting the serotonergic system result in body weight loss[26]. In addition, ablation studies have demonstrated that central serotonergic cells activate and recruit thermogenic brown and beige fat and regulate glucose and lipid homeostasis[27], and injection of orexin-A into RPa produces a sustained increase in brown adipose tissue (BAT) thermogenesis[28].

In this work, we systematically investigated the expression and distribution of orexin receptors in serotonergic neurons in RN, and the impact of optogenetic activation of orexin signaling and specific inactivation of Ox1R or Ox2R signaling in serotonin transporter (Sert)-expressing cells on glucose metabolism and energy homeostasis. The present study assigns orexin signaling in serotonergic neurons critical, yet differential orexin receptor type 1- and 2-dependent functions in the regulation of systemic glucose homeostasis.

## Results

**Distribution of Ox1R and Ox2R expression in serotonergic neurons in RN.** To systemically define the expression pattern of Ox1R and Ox2R in serotonergic neurons, Slc6a4-cre (Sert-Cre) mice were mated with tdTomato Flox (tdTomato fl/fl) mice to obtain Sert^tdTomato mice that express the red tomato (tdTomato) reporter protein upon Sert-Cre-mediated recombination of a loxP-flanked stop cassette[29] specifically in serotonergic neurons. Beyond the raphe nuclei, ectopic expression of tdTomato was also detected in deep layers of the cingulate cortex (Cg1 and Cg2), and ventral posterolateral (VPL), and ventral posteromedial (VPM) thalamic nuclei (Supplementary Fig. 1a–h). There were no detectable orexin receptors in tdTomato positive cells in these areas except for moderate expression levels of Ox2R in the cingulate cortex, as revealed by fluorescent in situ hybridization (RNAscope) employing probes directed to tdTomato, Ox1R (Hcrtr1), and Ox2R (Hcrtr2) mRNA (Supplementary Fig. 1i, k). Most of the Ox2R signal could not be detected in tdTomato positive cells in the cingulate cortex, where only five cells out of the 45 tdTomato positive cells (10.74%) expressed Ox2R (Supplementary Fig. 1j). As a positive control, abundant Ox1R and Ox2R mRNA was expressed in tdTomato positive neurons in DR (Supplementary Fig. 1l). We further confirmed the specificity of Cre expression in raphe nuclei of a Sert^tdTomato mouse. As markers for serotonergic neurons, tryptophan hydroxylase isoform 2 (TPH2) and Pet1 were co-expressed by 94.90% or all of analyzed cells in DR and RPa, respectively (Supplementary Fig. 1m, n, q). Here, almost all tdTomato positive neurons are TPH2 and/or Pet1 positive in DR, MR and RPa (Supplementary Fig. 1m, p, q). More than 95% of TPH2 and/or Pet1 positive neurons are tdTomato positive (Supplementary Fig. 1m, o, q, r). This indicates specific and efficient expression of Cre in serotonergic neurons in raphe nuclei.

The endogenous expression of Ox1R and Ox2R in serotonergic neurons in RN was systemically investigated in Sert^tdTomato mice. Ox1R was expressed in 66.01–89.20% of serotonergic neurons (tdTomato positive) in DR, MR, and RPa and the relative fluorescence signal in each neuron was strongest in DR-dorsal (DRD, ≈1.86) compared to other regions (≈0.64–0.92; Fig. 1a–d). In contrast, Ox2R was expressed in 63.43% of serotonergic neurons in DR-ventral (DRV), which was more than in DRD (≈45.36%), MR-dorsal (MRD, ≈28.18%), MR-ventral (MRV, ≈13.94%) and RPa (≈22.94%, Fig. 1a, b, e). The relative Ox2R fluorescence signal in each cell is also stronger in DRV (≈2.66) compared to DRD (≈1.33) and MRD (≈0.60, Fig. 1a, b, f). Of note, the Ox2R signal in serotonergic neurons in MRV (≈0.17) and RPa (≈0.24) was very weak (Fig. 1a, b, f).

Next, we analyzed published single-cell RNA sequencing (scRNA-seq) results of serotonergic neurons in DR[30,31]. 20.74% serotonergic neurons expressed Ox1R but not Ox2R (Ox1R-only), 6.25% serotonergic neurons expressed Ox2R but not Ox1R (Ox2R-only), only 2.27% serotonergic neurons expressed both orexin receptors (Ox1R/Ox2R+), and 70.74% serotonergic neurons expressed none of the orexin receptors (Ox1R/Ox2R−) (Fig. 1g, g'). Interestingly, Slc17a8 (vesicular glutamate transporter 3, VGLUT3) was the only gene showing significantly different expression levels, comparing Ox1R-only with Ox2R-only serotonergic neurons (Source Data of Fig. 1i). The average expression level of VGLUT3 was higher in Ox2R-only than Ox1R-only

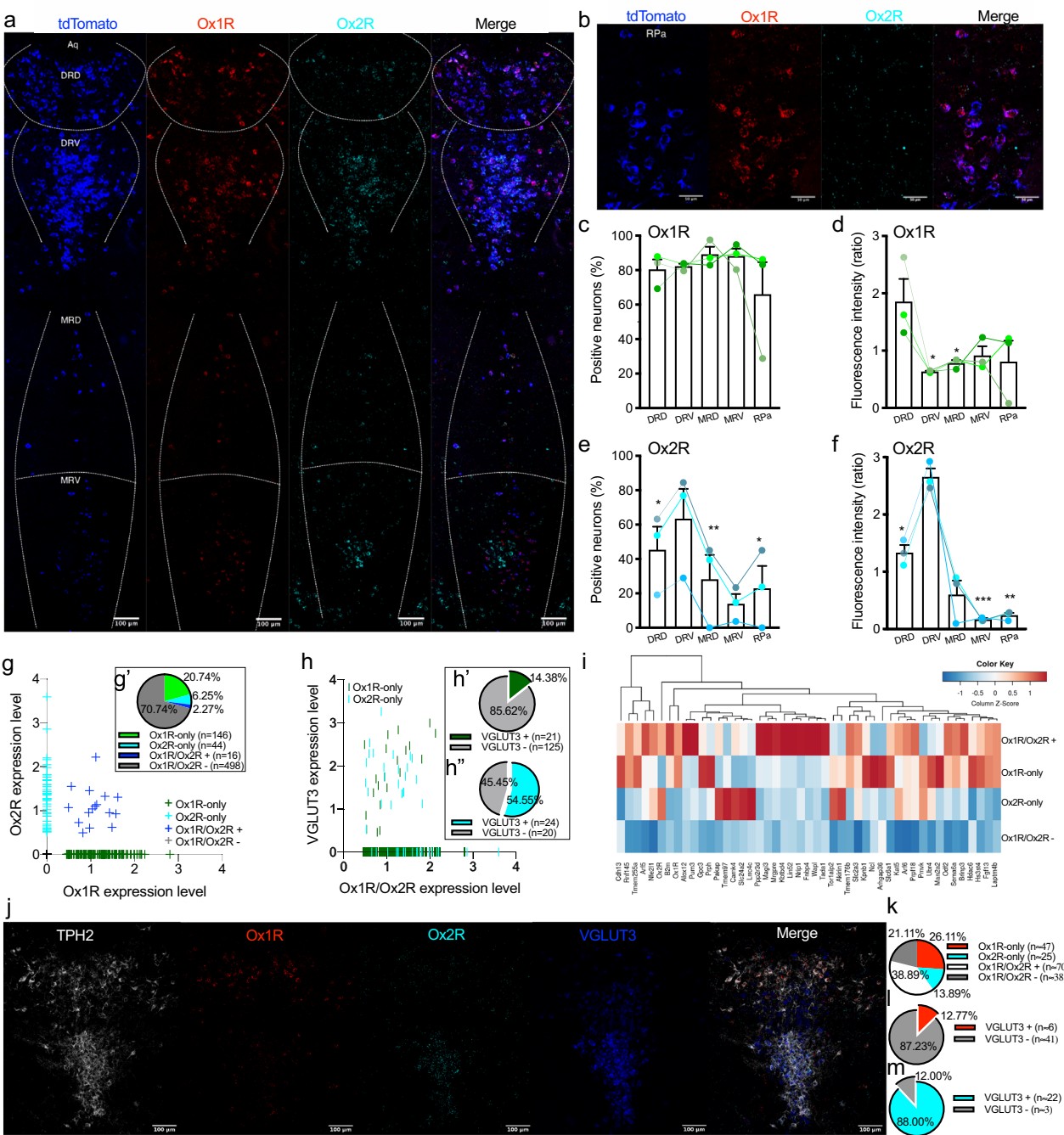

**Fig. 1 Distribution of Ox1R and Ox2R in serotonergic neurons of the raphe nucleus of a SERT<sup>tdTomato</sup> mouse. a** Representative images of RNAscope in situ hybridization in the dorsal raphe nucleus (dorsal (DRD), ventral (DRV)), and median raphe nucleus (dorsal (MRD), ventral (MRV)). **b** Representative images of RNAscope in situ hybridization in the raphe pallidus (RPa). **c, e** Percentages of Ox1R or Ox2R positive neurons in serotonergic neurons. **d, f** The average of relative fluorescence intensity of Ox1R or Ox2R signal in each serotonergic neuron. Fluorescence was normalized by the mean of all regions for each mouse. Blue, tdTomato; red, Ox1R; cyan, Ox2R. Scale bar: 100 µm in (**a**) and 50 µm in (**b**). $n = 3$. Data are represented as means ± SEM. **d** DRV: $p = 0.035$, MRD: $p = 0.036$; **e** DRD: $p = 0.049$, MRD: $p = 0.0081$, RPa: $p = 0.028$; **f** DRD: $p = 0.010$, MRV: $p = 0.0008$, RPa: $p = 0.0099$. *$p < 0.05$, **$p < 0.01$, ***$p < 0.001$; compared to DRD (**d**) or DRV (**e, f**), as determined by paired two-tailed $t$-test (**c, e**) or ratio paired two-tailed $t$-test (**d, f**). Analysis of published scRNA-Seq data of serotonergic neurons in dorsal raphe nucleus (DR)[30,31]: **g** Scatter plot of raw cell read counts of Ox1R and Ox2 R, **g'** percentages of serotonergic neurons only expressing Ox1R (Ox1R-only) or Ox2R (Ox2R-only) and those expressing both or none of Ox1R and Ox2R (Ox1/2R+ or Ox1/2 R−), **h** scatter plot of raw vesicular glutamate transporter 3 (VGLUT3) counts against raw Ox1R or Ox2R counts, **h'** percentages of VGLUT3 positive (+) and negative (−) neurons in Ox1R-only and **h″** Ox2R-only serotonergic neurons, and **i** heat map of gene counts for cells with column-wise z-scores. Mean gene abundance was calculated per gene (x-dimension) and group (y-dimension). Column-wise z-scoring (per gene) was applied to make expression visually comparable between groups independent of average gene abundance levels. **j** Representative images of RNAscope in situ hybridization in the dorsal raphe nucleus (scale bar: 100 µm), and analysis: **k** percentages of Ox1R-only, Ox2R-only, Ox1/2R+ and Ox1/2R− serotonergic neurons; percentages of VGLUT3 positive neurons in (**l**) Ox1R-only and (**m**) Ox2R-only serotonergic neurons. Gray, tryptophan hydroxylase isoform 2 (TPH2); red, Ox1R; cyan, Ox2R; blue, VGLUT3. $n = 3$. Source data are provided as a Source Data file.

serotonergic neurons, mainly due to the fact that more Ox2R-only serotonergic neurons are VGLUT3 positive (Fig. 1h,h',h", Source Data of Fig. 1i). Further, we compared the gene expression patterns among Ox1R-only, Ox2R-only, Ox1R/Ox2R+ (both positive), and Ox1R/Ox2R− (both negative) serotonergic neurons, and revealed 47 differently expressed genes (Fig. 1i, Source Data of Fig. 1i).

In scRNAseq analysis, the percentage of orexin-receptor-positive serotonergic neurons was lower than what we detected by RNAscope. Due to different sensitivity of detecting genes between scRNA-seq and RNAscope, Ox1R-only and Ox2R-only serotonergic neurons revealed by scRNA-seq could be similar to Ox1R- and Ox2R-dominant serotonergic neurons revealed by RNAscope. We further confirmed scRNAseq findings using RNAscope combined with TPH2 immunostaining, which revealed 26.11%, 13.89%, 38.89%, and 21.11% Ox1R-only, Ox2R-only, Ox1R/Ox2R+ and Ox1R/Ox2R− serotonergic cells, respectively (Fig. 1j, k). VGLUT3 was mainly expressed in DRV, and more Ox2R-only serotonergic neurons are VGLUT3 positive (88.00%) compared to Ox1R-only serotonergic neurons (12.77%) (Fig. 1j, l, m).

**Selective inactivation of Ox1R and Ox2R in serotonergic neurons in RN.** To specifically inactivate Ox1R or Ox2R in serotonergic neurons (Ox1R$^{\Delta SERT}$, Ox2R$^{\Delta SERT}$), Ox1R Flox (Ox1R fl/fl), and Ox2R Flox (Ox2R fl/fl) mice were generated via homologous recombinant targeting in embryonic stem (ES) cells (Supplementary Fig. 2). The resulting animals were crossed with Sert-Cre mice to obtain Ox1R fl/fl, Sert-Cre tg/wt mice (Ox1R$^{\Delta SERT}$), and Ox2R fl/fl, Sert-Cre tg/wt mice (Ox2R$^{\Delta SERT}$). Mice from the same breeding without the Cre transgene were used as controls (Ox1R fl/fl, Sert-Cre wt/wt or Ox2R fl/fl, Sert-Cre wt/wt mice). Ox1R$^{\Delta SERT}$ or Ox2R$^{\Delta SERT}$ mice were further intercrossed with Ox1R fl/fl, tdTomato fl/fl mice or Ox2R fl/fl, tdTomato fl/fl mice, respectively, to obtain Ox1R fl/fl, tdTomato fl/−, Sert-Cre tg/wt (Ox1R$^{\Delta SERT/tdTomato}$) mice or Ox2R fl/fl, tdTomato fl/−, Sert-Cre tg/wt (Ox2R$^{\Delta SERT/tdTomato}$) mice that express tdTomato but lack Ox1R or Ox2R in serotonergic neurons.

In Ox1R$^{\Delta SERT/tdTomato}$ or Ox2R$^{\Delta SERT/tdTomato}$ mice, we confirmed the specific deletion of Ox1R or Ox2R, respectively, in serotonergic neurons in RN via RNAscope detection of tdTomato, Ox1R, and Ox2R mRNA. In Ox1R$^{\Delta SERT/tdTomato}$ mice, the Ox1R signal was almost completely abrogated in serotonergic neurons and only single-labeled, non-Sert Ox1R-expressing cells were observed, while the Ox2R signal exhibited a similar expression pattern to that in the Sert$^{tdTomato}$ mice (Fig. 2a, c). In contrast, in Ox2R$^{\Delta SERT/tdTomato}$ mice, the Ox1R signal was similar to that in the Sert$^{tdTomato}$ mice, while the Ox2R signal almost disappeared in serotonergic neurons and some single-labeled, non-Sert Ox2R-expressing cells were observed (Fig. 2b, d). Accordingly, Ox1R$^{\Delta SERT/tdTomato}$ mice showed significantly lower percentages of Ox1R positive neurons in serotonergic neurons and also significantly lower raw fluorescence intensity of Ox1R signal in individual serotonergic neurons in all RN regions compared to Ox2R$^{\Delta SERT/tdTomato}$ mice (Fig. 2e, f). Ox2R$^{\Delta SERT/tdTomato}$ mice showed significantly lower percentages of Ox2R positive neurons in serotonergic neurons in DRD, DRV, and MRD and also significantly lower raw fluorescence intensity of Ox2R signal in individual serotonergic neurons in DRD and DRV than Ox1R$^{\Delta SERT/tdTomato}$ mice (Fig. 2g, h). In Ox2R$^{\Delta SERT/tdTomato}$ mice, Ox1R fluorescence intensity in each DRD serotonergic neuron was highest among all RN regions and significantly higher than that in DRV, MRD, and MRV (Fig. 2f). In Ox1R$^{\Delta SERT/tdTomato}$ mice, Ox2R mRNA expression levels in serotonergic neurons in DRV were highest and significantly higher than levels in MRD, MRV, and RPa, as quantified in percentages of positive neurons and raw fluorescent intensity (Fig. 2g, h).

Notably, in serotonergic neurons in MRV and RPa, Ox2R mRNA expression levels were similar between Ox1R$^{\Delta SERT/tdTomato}$ mice and Ox2R$^{\Delta SERT/tdTomato}$ mice, which indicates that this signal was close to the background fluorescence related to method limitations, presumably (Fig. 2g, h). Therefore, only Ox1R but not Ox2R was detectable in serotonergic neurons in MRV and RPa.

**Differential activation of DR serotonergic neurons by orexin.** To further validate the deletion of orexin receptor signaling, we performed perforated patch-clamp recordings and Ca$^{2+}$ imaging on DR serotonergic neurons from control, Ox1R$^{\Delta SERT}$, and Ox2R$^{\Delta SERT}$ mice with tdTomato reporter or genetically encoded calcium indicator GCaMP6 in serotonergic neurons (Fig. 3a–e). During the experiments, GABAergic and glutamatergic synaptic input was blocked.

In the first series of experiments we performed patch-clamp recordings in the DR with the focus mainly on the DRD. 100 nM orexin A increased the action potential firing rate in control serotonergic neurons ($p < 0.0001$, n = 21; Fig. 3b), which is in line with previous studies[32–34]. Similarly, 100 nM orexin A increased the activity of serotonergic neurons in Ox1R$^{\Delta SERT/tdTomato}$ ($p < 0.05$, n = 9; Fig. 3b) and Ox2R$^{\Delta SERT/tdTomato}$ mice ($p < 0.005$, n = 11; Fig. 3b). In contrast, 100 nM orexin B increased action potential firing rate only in control ($p < 0.0001$, n = 27; Fig. 3c) and Ox1R$^{\Delta SERT/tdTomato}$ mice ($p < 0.0001$, n = 14; Fig. 3c) but not in Ox2R$^{\Delta SERT/tdTomato}$ mice ($p = 0.28$, n = 11; Fig. 3c). A representative recording with response to 1, 10, and 100 nM of orexin A in a control neuron was shown in Supplementary Fig. 3a, b.

In the second series of experiments we focused on the analysis of serotonergic neurons in the DRV using Ca$^{2+}$ imaging with the genetically encoded calcium indicator GCaMP6 (Fig. 3d, e). 100 nM orexin A increased [Ca$^{2+}$]$_i$ in control ($p < 0.0001$, n = 159), Ox1R$^{\Delta SERT}$ ($p < 0.0001$, n = 200) and Ox2R$^{\Delta SERT}$ neurons ($p < 0.0001$, n = 75). The increase in [Ca$^{2+}$]$_i$ was significantly smaller in Ox2R$^{\Delta SERT}$ neurons compared to control ($p < 0.0001$) and Ox1R$^{\Delta SERT}$ ($p < 0.0001$) neurons. 100 nM orexin B increased [Ca$^{2+}$]$_i$ in control ($p < 0.0001$, n = 75) and Ox1R$^{\Delta SERT}$ neurons ($p < 0.0001$, n = 92) and had no effect on Ox2R$^{\Delta SERT}$ neurons ($p = 0.8$, n = 122).

The differential and complex response patterns are likely caused by the differences in affinity profiles of orexin A and orexin B to Ox1R and Ox2R[4], and the varying rate of Ox1R and Ox2R expression in different regions of the DR as described above. In radioligand binding assays, orexin-A had high affinities to Ox1R (IC$_{50}$ = 20 nM) as well as to Ox2R (IC$_{50}$ = 38 nM), whereas orexin B has a high affinity to Ox2R (IC$_{50}$ = 36 nM) and only a low affinity to Ox1R (IC$_{50}$ = 420 nM)[4]. While both orexin receptors are expressed in the DRD and the DRV, the relative expression of Ox1R is higher in the DRD, whereas the relative expression of Ox2R is higher in the DRV (Fig. 1a, d–f).

Since orexin-A is highly affine to both receptors, orexin A increases activity in serotonergic DRD neurons of Ox1R$^{\Delta SERT/tdTomato}$ and Ox2R$^{\Delta SERT/tdTomato}$ mice, even when one receptor is switched off. The effect of Ox1R deletion becomes only detectable at very low orexin A concentrations (Supplementary Fig. 3c) since the affinity of orexin A is higher to Ox1R than to Ox2R. Orexin B, in contrast, has only a low affinity for OxR1. Thus, when Ox2R are switched off, the neurons do not react anymore (Fig. 3c). In the DRV, the situation is different in the sense that Ox2R are relatively higher expressed than Ox1R. If Ox2R are missing, there are too few receptors to fully activate the serotonergic neurons of the DRV with orexin A. In the absence of Ox2R, orexin B can practically no longer trigger an effect due to its low affinity to Ox1R.

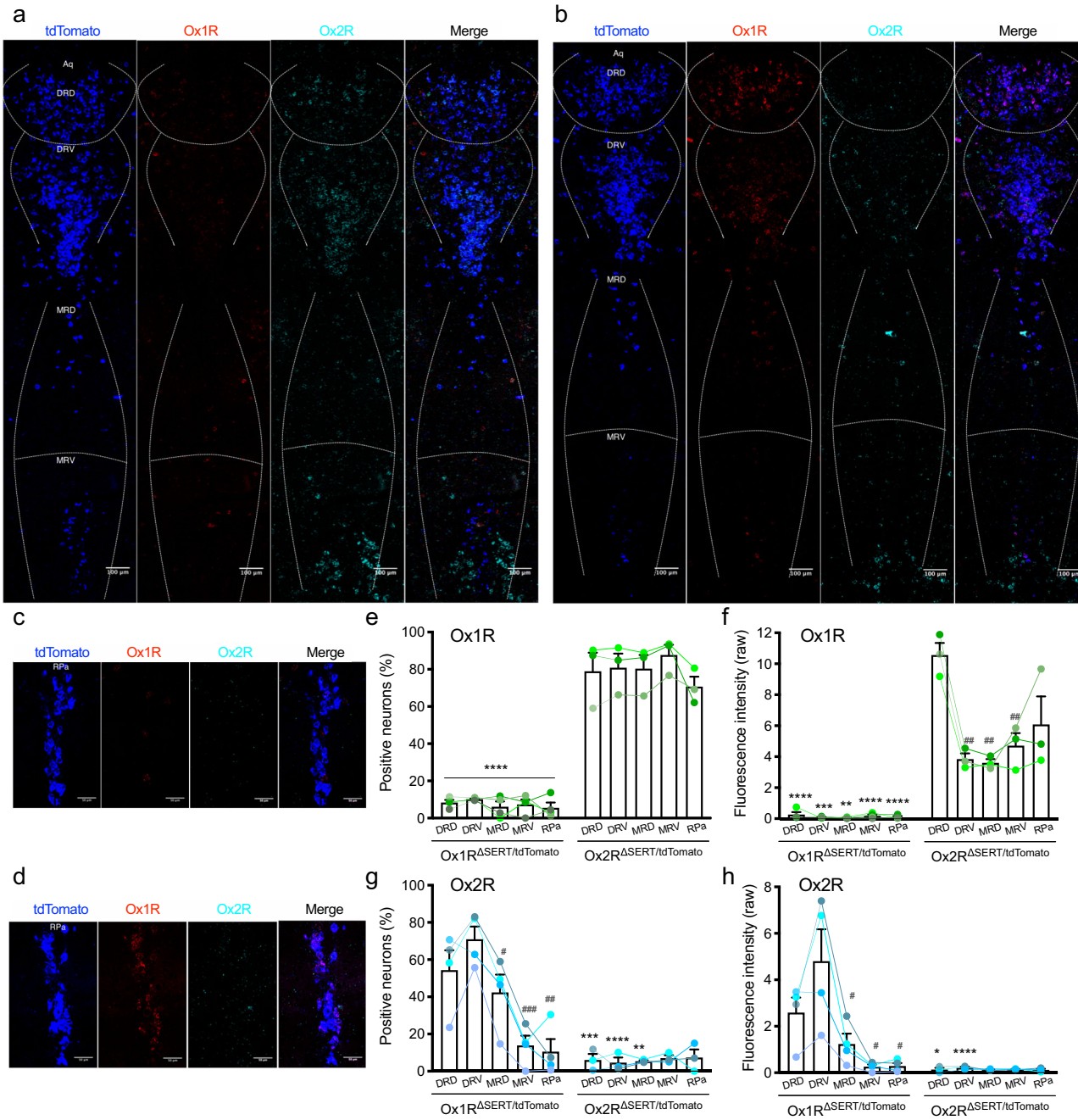

**Fig. 2 Specific inactivation of Ox1R or Ox2R in serotonergic neurons of a Ox1R$^{\Delta SERT/tdTomato}$ or Ox2R$^{\Delta SERT/tdTomato}$ mouse, respectively. a** Representative images of RNAscope in situ hybridization in the dorsal raphe nucleus (dorsal (DRD), ventral (DRV)) and median raphe nucleus (dorsal (MRD), ventral (MRV)) of a Ox1R$^{\Delta SERT/tdTomato}$ mouse and **b** a Ox2R$^{\Delta SERT/tdTomato}$ mouse. **c** Representative images of RNAscope in situ hybridization in the raphe pallidus (RPa) of a Ox1R$^{\Delta SERT/tdTomato}$ mouse and **d** a Ox2R$^{\Delta SERT/tdTomato}$ mouse. **e, g** Percentages of Ox1R or Ox2R positive neurons in serotonergic neurons. **f, h** The average of raw fluorescence intensity of Ox1R or Ox2R signal in each serotonergic neuron. Blue, tdTomato; red, Ox1R; cyan, Ox2R. Scale bar: 100 μm in (**a, b**) and 50 μm in (**c, d**). Ox1R$^{\Delta SERT/tdTomato}$, $n = 4$; Ox2R$^{\Delta SERT/tdTomato}$, $n = 3$. Data are represented as means ± SEM. **f** Ox1R$^{\Delta SERT/tdTomato}$: $p = 0.0005$ (DRV) and 0.0011 (MRD); Ox2R$^{\Delta SERT/tdTomato}$: $p = 0.0041$ (DRV), 0.0088 (MRD) and 0.0098 (MRV). **g** Ox1R$^{\Delta SERT/tdTomato}$: $p = 0.013$ (MRD), 0.0006 (MRV) and 0.0014 (RPa); Ox2R$^{\Delta SERT/tdTomato}$: $p = 0.0002$ (DRD) and 0.0046 (MRD). **h** Ox1R$^{\Delta SERT/tdTomato}$: $p = 0.038$ (MRD), 0.039 (MRV) and 0.037 (RPa); Ox2R$^{\Delta SERT/tdTomato}$: $p = 0.037$ (DRD). *$p < 0.05$, **$p < 0.01$, ***$p < 0.001$, ****$p < 0.0001$; as determined by two-way ANOVA followed by Sidak's post hoc test. The main effect of genotype: **e** ($F_{(1, 25)} = 583.60$, $p < 0.0001$), **f** ($F_{(1, 25)} = 234.20$, $p < 0.0001$), **g** ($F_{(1, 25)} = 53.99$, $p < 0.0001$), **h** ($F_{(1, 25)} = 20.08$, $p = 0.0001$). #$p < 0.05$, ##$p < 0.01$, ###$p < 0.001$, compared to DRD (**f**) or DRV (**g, h**), as determined by paired two-tailed $t$-test. Source data are provided as a Source Data file.

### Orexin neurons are activated in HFD-induced obesity.

In order to investigate the activation of orexin neurons in lean and obese mice, we compared c-Fos expression in orexin neurons in 6 h-fasted BL/6 mice, which had been fed a HFD for 10 weeks since weaning, to that in mice exposed for the same period to a control diet (CD). To this end, we determined the expression of orexin (*Hcrt*) and c-Fos (*Fos*) mRNA via RNAscope hybridization. This analysis revealed, that the proportion of c-Fos-expressing orexin neurons in the LH was

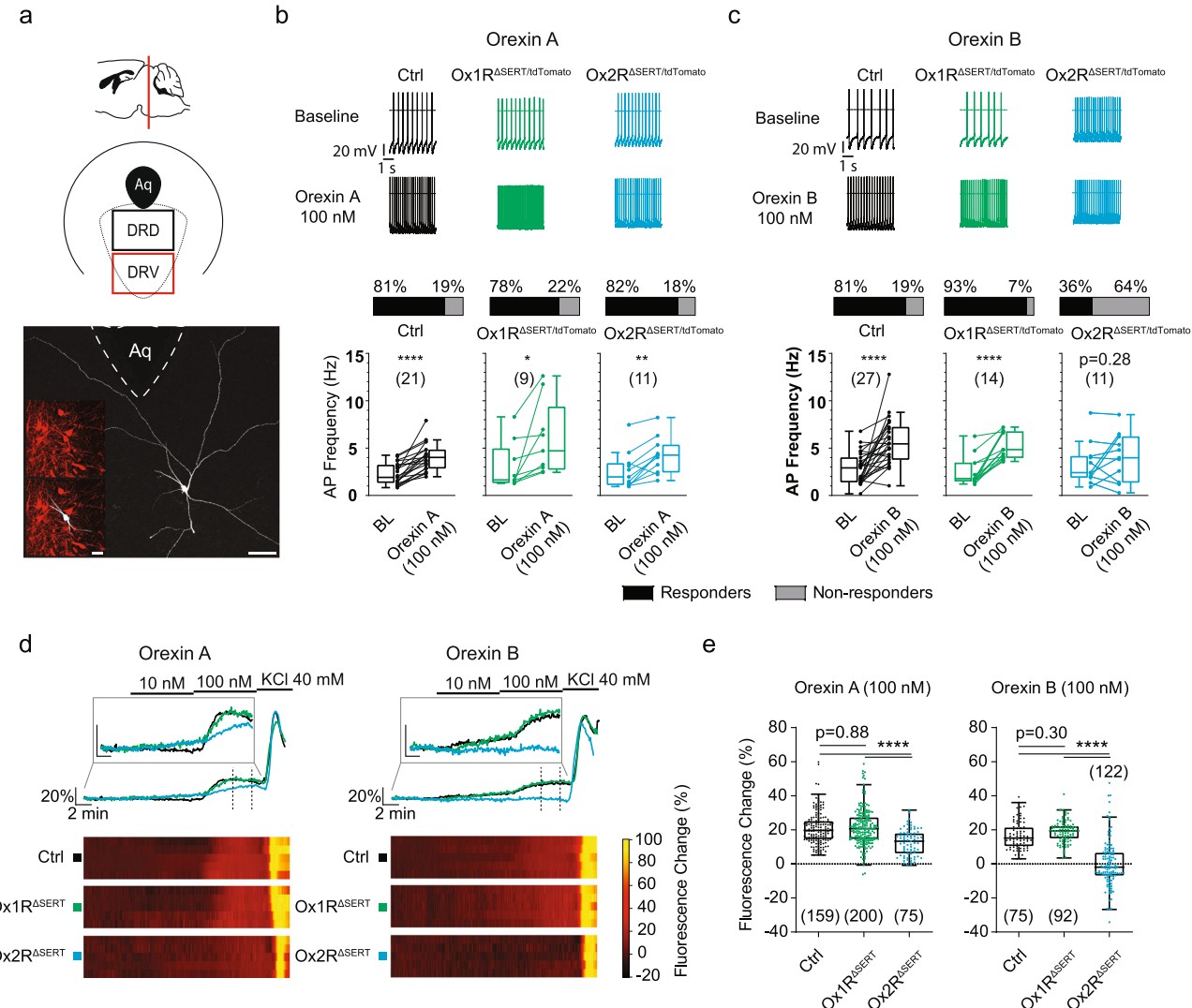

**Fig. 3 Effect of orexin A and orexin B on serotonergic neurons in the dorsal raphe nucleus (DR) analyzed by patch-clamp recordings and $Ca^{2+}$ imaging with GCaMP6.** Recordings were performed in brain slices from control, $Ox1R^{\Delta SERT}$, and $Ox2R^{\Delta SERT}$ male mice, with tdTomato or GCaMP6 expression in serotonergic neurons for patch clamp or $Ca^{2+}$ imaging, respectively. **a** Schematic illustration of the dorsal raphe nucleus (dorsal (DRD), ventral (DRV)) and an exemplary image of a recorded serotonergic neuron in DRD, which was biocytin-streptavidin labeled during the experiment. Scale bar: 50 μm. Insert: tdTomato labeling (top) and double labeling with tdTomato and biocytin-streptavidin (bottom). Scale bar: 20 μm. **b, c** Orexin A (**b**) and orexin B (**c**) effect on action potential firing rate of serotonergic neurons in DRD. Original recordings (top) and comparisons of mean firing rates (bottom). The stacked bars show the percentage of individual neurons in which the increase in action potential frequency was larger than 3 × SD of the control, thus defining them as responsive (see "Methods"). Mean firing rates were compared using paired two-tailed students $t$-test. **b** Orexin A application: Ctrl, $p < 0.0001$, $n = 21$; $Ox1R^{\Delta SERT}$: $p = 0.03$, $n = 9$; $Ox2R^{\Delta SERT}$: $p = 0.0018$, $n = 11$. **c** Orexin B application: Ctrl, $p < 0.0001$, $n = 27$; $Ox1R^{\Delta SERT}$: $p < 0.0001$, $n = 14$; $Ox2R^{\Delta SERT}$: $p = 0.28$, $n = 11$. Abbreviation: baseline (BL). **d, e** Orexin A and orexin B effect on $[Ca^{2+}]_i$ of serotonergic neurons in DRV measured with GCaMP6. **d** Original recordings (top) and heat maps of five individual neurons for each set of experiments (bottom). The recordings show the responses to the orexins and high $K^+$ saline. Dashed lines indicate the range where the responses were quantified. **e** Calcium responses upon 100 nM orexin A and orexin B. Data are shown as the percentage of the maximal response to high $K^+$ saline. Mean increases in $[Ca^{2+}]_i$ between experimental groups were compared, performing ANOVA with post hoc Tukey tests. Orexin A application: Ctrl vs $Ox1R^{\Delta SERT}$, $p = 0.88$; Ctrl vs $Ox2R^{\Delta SERT}$, $p < 0.0001$; $Ox1R^{\Delta SERT}$ vs $Ox2R^{\Delta SERT}$, $p < 0.0001$ (Ctrl, $n = 159$; $Ox1R^{\Delta SERT}$, $n = 200$; $Ox2R^{\Delta SERT}$, $n = 75$). Orexin B application: Ctrl vs $Ox1R^{\Delta SERT}$, $p = 0.30$; Ctrl vs $Ox2R^{\Delta SERT}$, $p < 0.0001$; $Ox1R^{\Delta SERT}$ vs $Ox2R^{\Delta SERT}$, $p < 0.0001$ (Ctrl, $n = 75$; $Ox1R^{\Delta SERT}$, $n = 92$; $Ox2R^{\Delta SERT}$, $n = 122$). In the box plots, the horizontal lines show the median of the data. Boxes indicate the 25th and 75th percentile. The whiskers were calculated according to the 'Tukey' method. *$p < 0.05$, **$p < 0.01$, ***$p < 0.001$, ****$p < 0.0001$. $n$ values are given in brackets. Source data are provided as a Source Data file.

significantly increased in mice exposed to HFD compared to CD feeding (Fig. 4a, b). Collectively, these data indicate an increased activation of orexin neurons in obese mice.

Then, we investigated the impact of Ox1R or Ox2R inactivation in serotonergic system on energy homeostasis under both normal chow diet (NCD)-fed lean condition and HFD-induced obese condition. There were no differences in body weight (BW; Fig. 4c, g), glucose tolerance under 16-h fasting condition (Fig. 4d, h) or insulin sensitivity under randomly fed condition (Fig. 4f, j) between control and $Ox1R^{\Delta SERT}$ mice. Two-way ANOVA revealed a significant decrease of glucose tolerance in obese ($F_{(1, 105)} = 7.86$, $p = 0.0060$) but not lean $Ox1R^{\Delta SERT}$ mice, compared to control mice after a 6-h fasting period, though there was no significant change at individual time points by post

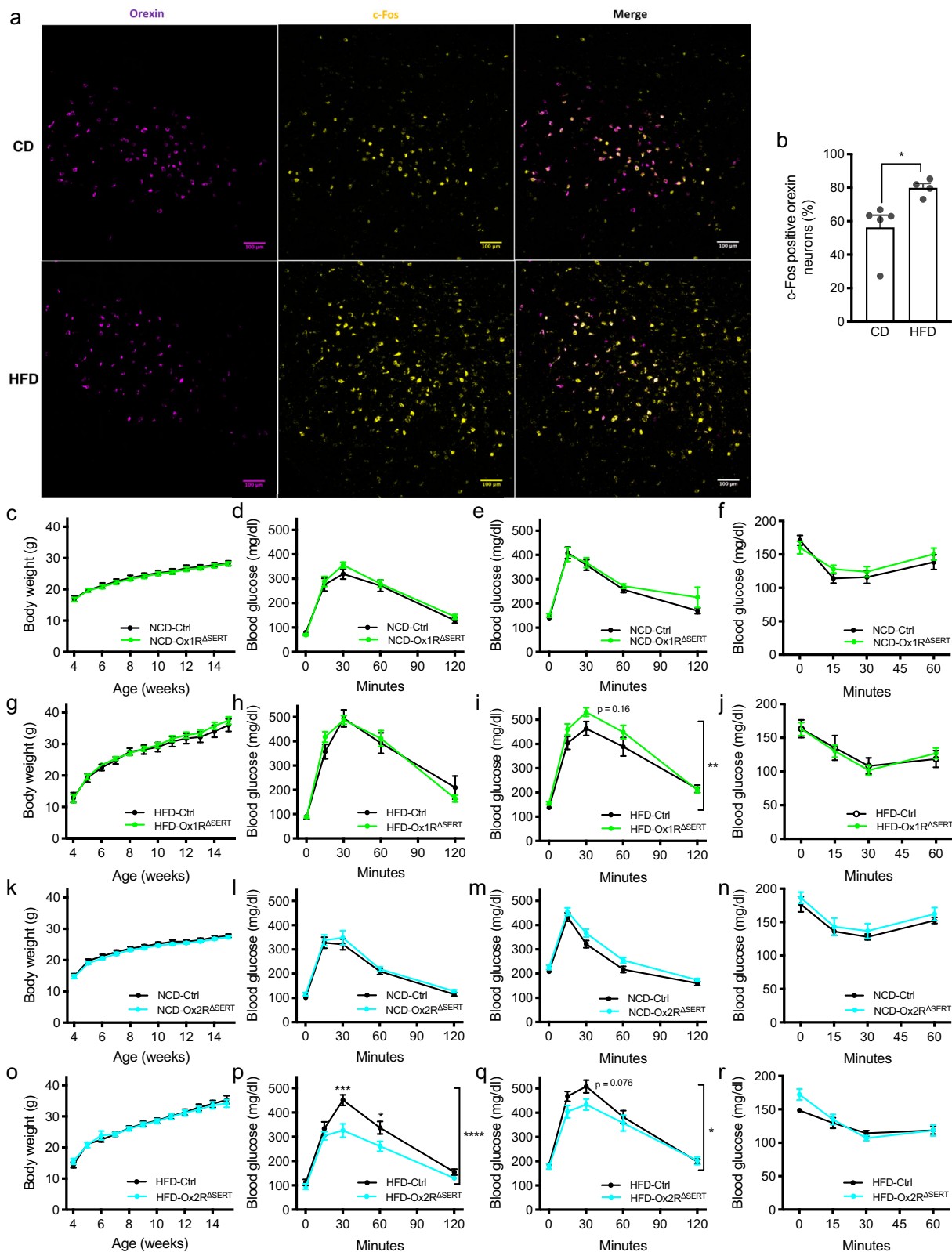

hoc test (Fig. 4e, i). Similarly, in lean Ox2R$^{\Delta SERT}$ mice, BW, glucose tolerance, and insulin sensitivity remained unchanged (Fig. 4k–n). Interestingly, in obese Ox2R$^{\Delta SERT}$ mice, glucose tolerance was significantly improved after a 16-h fasting period (Fig. 4p), while BW and insulin sensitivity remained unchanged, compared to control mice (Fig. 4o, r). Two-way ANOVA revealed a significant improvement of glucose tolerance in obese

Ox2R$^{\Delta SERT}$ mice, compared to control mice after a 6-h fasting period, though there was no significant change at individual time points by post hoc test (($F$ (1, 100) = 6.08, $p$ = 0.015), Fig. 4q).

Indirect calorimetry revealed unaltered locomotor activity and daily energy intake in Ox1R$^{\Delta SERT}$ and Ox2R$^{\Delta SERT}$ mice, compared to the respective control mice, under both lean and obese conditions (Supplementary Fig. 4). Collectively, these

**Fig. 4 Increased c-Fos activation in orexin neurons upon high-fat diet (HFD) feeding and improved glucose tolerance in Ox2R$^{\Delta SERT}$ mice fed a HFD. a** Representative images of RNAscope in situ hybridization in the lateral hypothalamus (LH) of a BL/6 mouse fed a control diet (CD) or HFD. $n = 5$ (CD) and 4 (HFD). **b** Percentages of c-Fos positive neurons in orexin neurons. $n = 5$ (CD) and 4 (HFD). $p = 0.029$. **c, g** Average body weight, glucose tolerance test under **d, h** 16-h and **e, i** 6-h fasting conditions, and **f, j** insulin tolerance test of control (Ctrl) and Ox1R$^{\Delta SERT}$ mice on normal chow diet (NCD) or HFD. NCD-Ctrl and NCD-Ox1R$^{\Delta SERT}$, $n = 9$; HFD-Ctrl, $n = 8$ (**g, h, j**) or 11 (**i**); HFD-Ox1R$^{\Delta SERT}$, $n = 10$ (**g, h, j**) or 12 (**i**). **i** $p = 0.16$ at 30 min. **k, o** Average body weight, glucose tolerance test under **l, p** 16-h and **m, q** 6-h fasting conditions, and **n, r** insulin tolerance test of Ctrl and Ox2R$^{\Delta SERT}$ mice on NCD or HFD. NCD-Ctrl, $n = 11$; NCD-Ox2R$^{\Delta SERT}$, $n = 8$; HFD-Ctrl, $n = 11$ (**o, p, r**) or 13 (**q**); HFD-Ox2R$^{\Delta SERT}$, $n = 9$. **p** $p = 0.0002$ (30 min) and 0.045 (60 min); **q** $p = 0.076$ (30 min). Magenta, orexin; yellow, c-Fos. Scale bar: 100 μm. Data are represented as means ± SEM. *$p < 0.05$, **$p < 0.01$, ***$p < 0.001$; as determined by unpaired two-tailed Student's $t$-test (**b**) or two-way ANOVA followed by Sidak's post hoc test (**i, p, q**). Two-way ANOVA revealed a significant main effect of genotype in (**i**) ($F_{(1, 105)} = 7.86$, $p = 0.0060$), **p** ($F_{(1, 90)} = 18.22$, $p < 0.0001$) and **q** ($F_{(1, 100)} = 6.08$, $p = 0.015$). Source data are provided as a Source Data file.

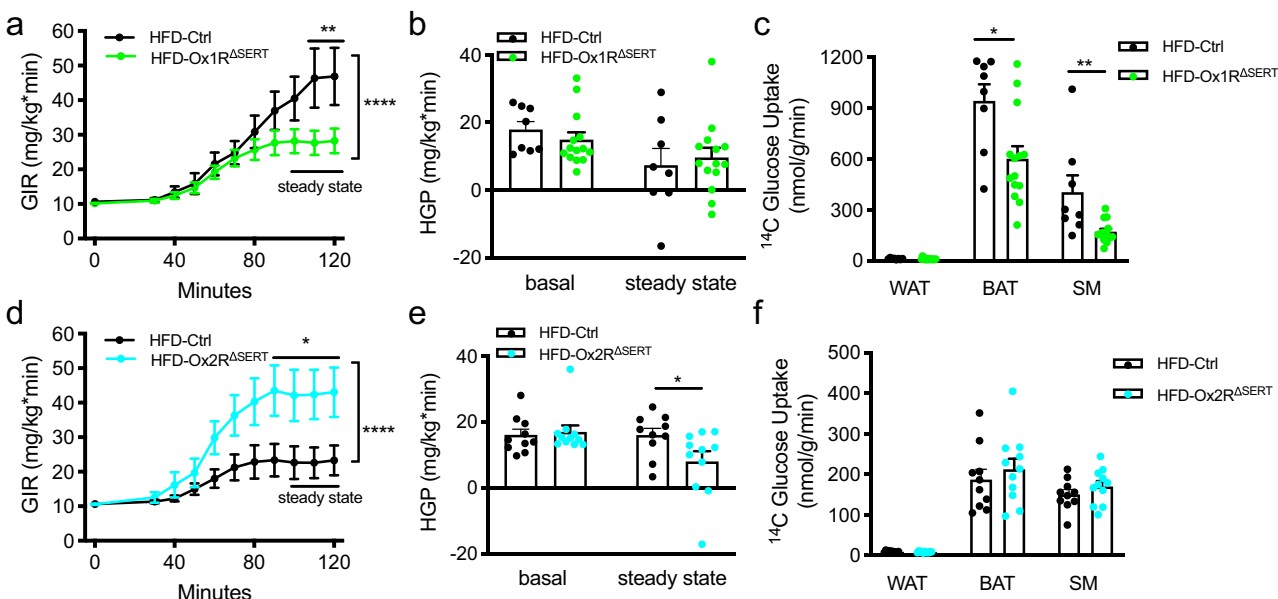

**Fig. 5 Insulin sensitivity is impaired in Ox1R$^{\Delta SERT}$ mice but improved in Ox2R$^{\Delta SERT}$ mice fed a HFD in hyperinsulinemic-euglycemic clamp analysis. a** Clamp glucose infusion rates (GIR), **b** hepatic glucose production (HGP) in the basal state and during the steady state of clamp analysis, and **c** tissue-specific insulin-stimulated [1-$^{14}$C]-Deoxy-D-glucose uptake in white adipose tissue (WAT), brown adipose tissue (BAT), and skeletal muscle (SM) under steady-state conditions of control (Ctrl) and Ox1R$^{\Delta SERT}$ mice. HFD-Ctrl, $n = 8$; HFD-Ox1R$^{\Delta SERT}$, $n = 14$. **a** $p = 0.0025$ (110 min) and 0.0028 (120 min); **c** $p = 0.011$ (BAT) and 0.0070 (SM). **d** Clamp GIR, **e** HGP in the basal state and during the steady state of clamp analysis, and **f** tissue-specific insulin-stimulated [1-$^{14}$C]-Deoxy-D-glucose uptake in WAT, BAT, and SM under steady-state conditions of control (Ctrl) and Ox2R$^{\Delta SERT}$ mice. HFD-Ctrl, $n = 10$; HFD-Ox2R$^{\Delta SERT}$, $n = 11$. **d** $p = 0.034$ (90 min), 0.047 (100 min), 0.041 (110 min) and 0.042 (120 min); **e** $p = 0.048$. Data are represented as means ± SEM. *$p < 0.05$, **$p < 0.01$; as determined by two-way ANOVA followed by Sidak's post hoc test (**a, d**) or unpaired two-tailed Student's $t$-test (**c, e**). Two-way ANOVA revealed a significant main effect of genotype in (**a**) ($F_{(1, 220)} = 18.30$, $p < 0.0001$) and (**d**) ($F_{(1, 209)} = 35.39$, $p < 0.0001$). Source data are provided as a Source Data file.

results indicate that glucose metabolism is improved in obese mice upon selective inactivation of Ox2R in serotonergic neurons.

**Insulin-stimulated HGP suppression and tissue glucose uptake in obese Ox1R$^{\Delta SERT}$ and Ox2R$^{\Delta SERT}$ mice.** To further specifically address the regulation of glucose metabolism in HFD-induced obese Ox1R$^{\Delta SERT}$ and Ox2R$^{\Delta SERT}$ mice, we performed hyperinsulinemic-euglycemic clamp studies. The glucose infusion rate (GIR) required to maintain euglycemia during steady-state conditions was significantly reduced in Ox1R$^{\Delta SERT}$ mice, compared to their littermate control mice (Fig. 5a, Supplementary Fig. 5a). Hepatic glucose production (HGP) was not significantly different between control and Ox1R$^{\Delta SERT}$ mice under basal clamp conditions or in response to insulin (Fig. 5b). Interestingly, the rate of plasma glucose disappearance (Rd) was significantly decreased in Ox1R$^{\Delta SERT}$ mice at a steady state while the levels under basal clamp conditions were not significantly different (Supplementary Fig. 5b). Determination of tissue-specific glucose uptake revealed impaired glucose uptake into

BAT and skeletal muscle (SM) of obese Ox1R$^{\Delta SERT}$ mice, while glucose uptake into white adipose tissue (WAT) remained unchanged (Fig. 5c). In contrast, GIR was significantly increased during steady-state conditions in Ox2R$^{\Delta SERT}$ mice compared to their littermate control mice (Fig. 5d, Supplementary Fig. 5d). Ox2R$^{\Delta SERT}$ mice exhibited similar HGP levels under basal clamp conditions but significantly lower HGP at steady state, compared to control mice, indicating that the suppression of HGP was more efficient in response to insulin in obese Ox2R$^{\Delta SERT}$ mice (Fig. 5e). There was no significant difference of Rd or tissue-specific glucose uptake between control and Ox2R$^{\Delta SERT}$ mice (Fig. 5f, Supplementary Fig. 5e). The insulin infusion was confirmed by detecting human insulin levels in serum at baseline and at the end of hyperinsulinemic-euglycemic clamp studies (Supplementary Fig 5c, f).

**Altered liver and BAT in obese Ox1R$^{\Delta SERT}$ and Ox2R$^{\Delta SERT}$ mice in hyperinsulinemic-euglycemic clamp studies.** To investigate the mechanisms underlying the effect of serotonergic-

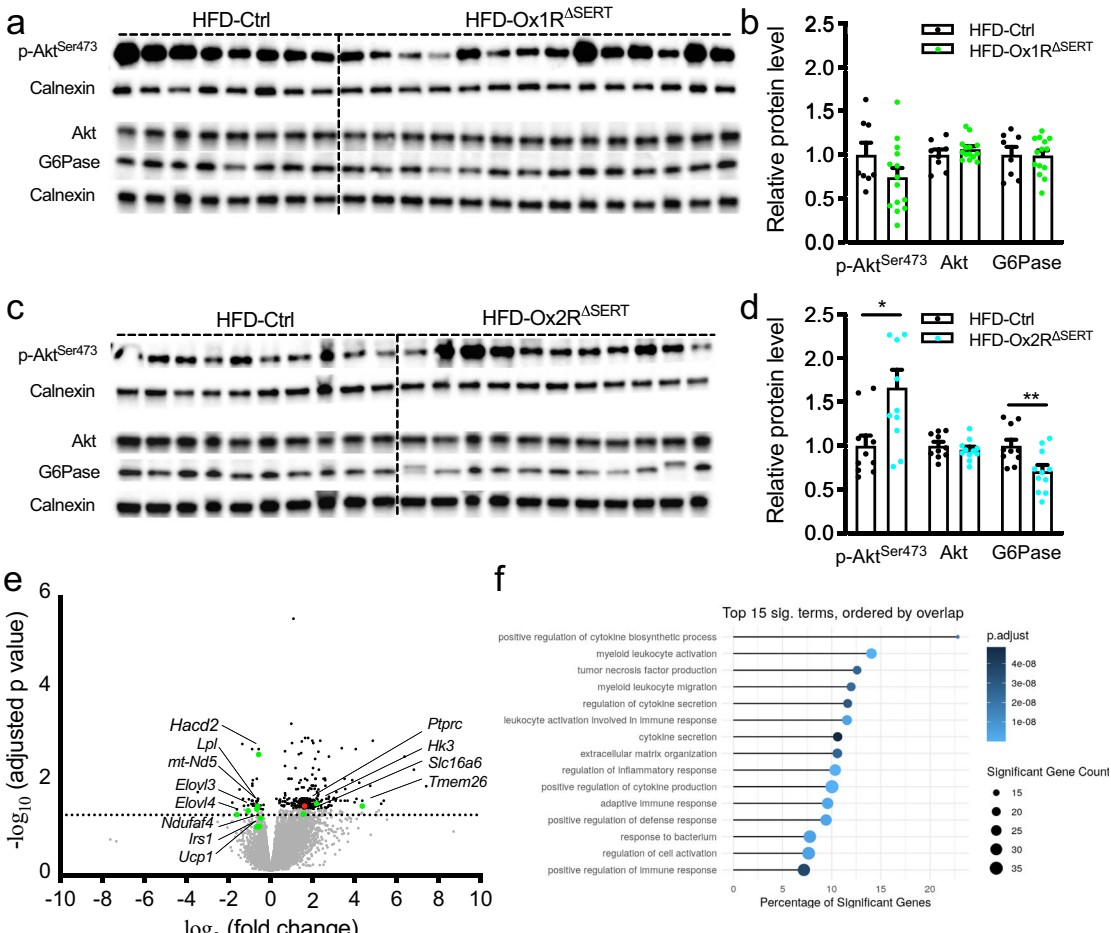

**Fig. 6 Insulin signaling in liver and BAT in obese Ox1R$^{\Delta SERT}$ and Ox2R$^{\Delta SERT}$ mice. a**, **b** Western blot images and quantification of p-Akt$^{Ser473}$, Akt, and G6Pase protein in the liver of control and Ox1R$^{\Delta SERT}$ mice. HFD-Ctrl, $n = 8$; HFD-Ox1R$^{\Delta SERT}$, $n = 14$. **c**, **d** Western blot images and quantification of p-Akt$^{Ser473}$, Akt, and G6Pase protein in the liver of control and Ox2R$^{\Delta SERT}$ mice. HFD-Ctrl, $n = 10$; HFD-Ox2R$^{\Delta SERT}$, $n = 11$. $p = 0.013$ (p-Akt$^{Ser473}$) and 0.0085 (G6Pase). Data are represented as means ± SEM. *$p < 0.05$, **$p < 0.01$; as determined by unpaired two-tailed Student's $t$-test. **e** Volcano plot of differential expression analysis of RNA sequencing of BAT in Ox1R$^{\Delta SERT}$ mice, compared to control mice. Some genes of interest are annotated. The differential gene expression test was done using negative binomial generalized linear models implemented in DESeq2 1.26.0. $P$-values are false discovery rates adjusted using the Benjamini-Hochberg procedure. **f** Top 15 differentially regulated gene ontology (GO) terms of class biological process in BAT of Ox1R$^{\Delta SERT}$ mice, compared to control mice. Significance is mapped to color, the dot size represents the number of significant genes in the GO term and the $x$-axis maps the percentage of significant genes to the overall gene GO term size. Gene-ontology term analysis of the 265 differentially expressed genes was carried out using the clusterProfiler R package, which utilizes an over-representation analysis calculating $p$-values by hypergeomtric distributions. $P$-values are FDR-adjusted. $n = 4$. Source data are provided as a Source Data file.

specific inactivation of Ox1R or Ox2R in hyperinsulinemic-euglycemic clamp studies, we further analyzed gene and protein expression in the liver and BAT dissected from these animals under clamp steady-state conditions.

In the liver, p-Akt$^{Ser473}$ protein, the major downstream effector of the insulin signaling pathway, was significantly increased in Ox2R$^{\Delta SERT}$ mice but unchanged in Ox1R$^{\Delta SERT}$ mice, compared to control mice, while the total Akt protein levels remained unchanged (Fig. 6a–d). Glucose-6-phosphatase (G6Pase) protein, which is essential for gluconeogenesis, was significantly decreased in Ox2R$^{\Delta SERT}$ mice but unchanged in Ox1R$^{\Delta SERT}$ mice, compared to control mice (Fig. 6a–d).

To investigate the mechanisms of glucose uptake reduction in BAT of obese Ox1R$^{\Delta SERT}$ mice compared to control mice, we compared the gene expression profiles of BAT dissected under clamp steady-state conditions from Ox1R$^{\Delta SERT}$ and control mice by total mRNA sequencing. This analysis revealed 265 genes significantly regulated in Ox1R$^{\Delta SERT}$ mice compared to control mice. The fatty acid elongase genes *Elovl3* and *Elovl4*, the

lipoprotein lipase gene *Lpl*, the very-long-chain (3R)-3-hydroxyacyl-CoA dehydratase 2 gene *Hacd2*, the mitochondrial genes *mt-Nd5* and *Ndufaf4* involved in Complex I and oxidative phosphorylation, the uncoupling protein 1 gene *Ucp1* and the insulin receptor substrate 1 gene *Irs1* were decreased in Ox1R$^{\Delta SERT}$ mice compared to control mice, though some of them were statistically non-significant (Fig. 6e). In addition, the beige adipocytes marker transmembrane marker 26 gene *Tmem26* and genes regulating glucose metabolism, including the solute carrier family 16 member 6 gene *Slc16a6* and hexokinase 3 gene *Hk3*, were increased in expression (Fig. 6e). Of note, the leukocyte antigen CD45 gene *Ptprc* was significantly increased and the ontology enrichment analysis revealed robust increase of immune response and inflammation in Ox1R$^{\Delta SERT}$ mice compared to control mice (Fig. 6e, f). The top 15 enriched gene ontology (GO) terms include positive regulation of cytokine production, myeloid leukocyte activation, regulation of inflammatory response, etc. (Fig. 6f). Among the significantly regulated genes, numerous relate to immune system, such as *Cd5l*, *Fcgr1*,

*Trem2, CD80, Tnfsf8, Tnfaip6, Cxcl3, Ccr1* (Source Data of Fig. 6e).

**Impact on BAT in obese Ox1R$^{\Delta SERT}$ and Ox2R$^{\Delta SERT}$ mice**. Next, we assessed BAT weight and morphology in HFD-induced obese Ox1R$^{\Delta SERT}$ mice. The percentage of BAT mass weight to BW was significantly increased in Ox1R$^{\Delta SERT}$ mice compared to control mice (Supplementary Fig. 6a). Histomorphological analysis revealed large lipid droplets in BAT of Ox1R$^{\Delta SERT}$ mice (Fig. 7a) and transmission electron microscopy (TEM)-analyses revealed that the aspect ratio (the ratio of long axis to the short axis) of mitochondria was significantly decreased in Ox1R$^{\Delta SERT}$ mice, while the average mitochondrial area remained unchanged (Fig. 7b–d). Analysis of relative mitochondrial DNA content revealed no changes between control and Ox1R$^{\Delta SERT}$ mice (Fig. 7e). Gene expression analyses demonstrated reduced expression of fatty acid oxidation-associated enzyme *Acox3*; *Vegfa*, which modulates mitochondrial function; *Ppargc1a*, which is important for mitochondrial biogenesis; *Cycs*, which functions in oxidative phosphorylation (OXPHOS); and *Dnm1l*, which mediates mitochondrial fission process in Ox1R$^{\Delta SERT}$ mice compared to their littermate controls (Fig. 7f, Supplementary Table 1). *Mfn1* gene, which mediates mitochondrial fusion showed a tendency to be decreased in Ox1R$^{\Delta SERT}$ mice, while *Mfn2*, *Fis1,* and *Mff* genes mediating mitochondrial fusion or fission were not significantly changed (Fig. 7f, Supplementary Table 1). Protein expression analysis supported the findings of impaired mitochondrial dynamics, oxidative phosphorylation, and thermogenesis. In Ox1R$^{\Delta SERT}$ mice, the protein expression of UCP-1, OXPHOS complex I CI-NDUFB8, complex IV CIV-MTCO1, mitofusin 1 (MFN1), and fission 1 (FIS1) were significantly reduced and complex II CII-SDHB also showed a tendency to be decreased (Fig. 7g–l). The mitochondrial protein import receptor TOM20, complex III CIII-UQCRC2, complex V CV-ATP5A, fusion proteins optic atrophy type 1 (OPA1) and mitofusin 2 (MFN2), and fission proteins dynamin-related protein 1 (DRP1) and mitochondrial fission factor (MFF) remained unchanged (Fig. 7g–l).

In contrast to obese Ox1R$^{\Delta SERT}$ mice, obese Ox2R$^{\Delta SERT}$ mice showed unaltered BAT mass weight, morphology as detected with haematoxylin and eosin (H&E) staining, gene expression levels of *Acox3*, *Vegfa*, *Ppargc1a*, *Cycs*, *Mfn1*, *Mfn2*, *Dnm1l*, *Fis1* and *Mff*, and protein expression levels of UCP1, compared to control mice (Fig. 7m–p, Supplementary Fig. 6b).

**Glucose metabolism upon optogenetic stimulation of orexin neurons in LH or orexinergic fibers in RPa**. Next, we investigated the impact of activation of the orexin system on glucose metabolism by insulin tolerance test (ITT) and glucose tolerance test (GTT), using an optogenetic approach. Orexin-Cre mice were created and the expression pattern of Cre was analyzed by crossing Orexin-Cre transgenic mice and channelrhodopsin-2 (ChR2)-tdTomato fl/fl mice. tdTomato was specifically expressed in orexin A-positive cells in LH (Supplementary Fig. 7e, k). In addition, we could detect a small cluster of tdTomato single-labeled cells below the lateral ventricle and around the 3rd ventricle and several scattered cells in other brain regions (Supplementary Fig. 7a–j). We injected an adeno-associated virus (pAAV-EF1a-double floxed-hChR2-EYFP) into the LH of Orexin-Cre mice to obtain expression of ChR2 and enhanced yellow fluorescent protein (EYFP) induced by Cre-loxP recombination in orexin neurons (Orexin$^{ChR2-EYFP}$). Control mice were injected with pAAV-Ef1a-DIO-EYFP (Orexin$^{EYFP}$). The efficiency and specificity of ChR2-EYFP and EYFP expression were evaluated with immunostaining of orexin and EYFP in LH. The

EYFP signal could be localized in orexin-positive cells exclusively (Fig. 8a). 70.95% and 76.23% of orexin neurons were EYFP positive in Orexin$^{EYFP}$ and Orexin$^{ChR2-EYFP}$ mice, respectively (Fig. 8b).

The optical fiber was implanted above LH in AAV-injected mice to allow optogenetic activation of orexin neurons (Fig. 8c, Supplementary Fig. 8a). Blue light (473 nm) laser illumination (20 mW, 20 Hz, 10 ms pulse, 10 s on in every 45 s), which started 30 min before glucose or insulin injection and lasted until the end of ITT and GTT experiments under random-fed or 6-h fasting conditions respectively, significantly impaired glucose tolerance 15 min after glucose injection (Fig. 8d) and insulin sensitivity revealed by two-way ANOVA ($F$ (1, 80) = 7.00, $p = 0.0098$; Fig. 8e). Accordingly, c-Fos expression was significantly increased in orexin neurons of Orexin$^{ChR2-EYFP}$ mice compared to Orexin$^{EYFP}$ mice (fasted for 6 h) after 1 h of laser illumination (Fig. 8f, g). c-Fos positive orexin neurons were detectable throughout the LH and we did not observe topographic segregation. DR- and RPa- projecting cells were distinct and intermingled in LH, as revealed by retrograde tracing with red and green retrobeads injected to DR and RPa respectively (Fig. 8h–j).

Innervation of orexinergic nerve fibers was analyzed by staining of serotonin, orexin, and EYFP in RN of AAV-injected mice. There were abundant orexin- and EYFP-positive nerve fibers which co-localized with serotonin neurons in all RN regions, including DRD, DRV, MRD, MRV, and RPa, of control (Fig. 9a) and Orexin$^{ChR2-EYFP}$ mice (Fig. 9b). This indicates that orexin nerve fibers innervate RN and the ChR2 protein travels successfully to these nerve endings. Further, we implanted the optical fiber above RPa in AAV-injected mice to selectively activate orexinergic fibers in this projection field (Fig. 9c, Supplementary Fig. 8b). Laser illumination (10 mW, 20 Hz, 10 ms pulse, 10 s on in every 45 s) in RPa, which started 30 min before glucose or insulin injection and lasted until the end of experiments, significantly improved glucose tolerance 15 min after glucose injection in GTT under 6-h fasting conditions (Fig. 9d). Insulin sensitivity remained unchanged upon laser illumination in RPa, as assessed by ITT (Fig. 9e). c-Fos positive serotonergic neurons in RPa were significantly increased in Orexin$^{ChR2-EYFP}$ mice compared to Orexin$^{EYFP}$ mice (fasted for 6 h) after 1 h of laser illumination (Fig. 9f, g), without evidence for significant back-propagation to orexin neurons in the LH (Supplementary Fig. 9e, f).

Laser illumination in LH or RPa failed to affect glucose metabolism in control mice, as measured by ITT and GTT (Supplementary Fig. 9a–d).

## Discussion
DR and MR in the midbrain and pons contain the majority of serotonergic neurons, which mainly project to the forebrain, while serotonergic neurons in caudal RN, such as RPa, mainly project to the brain stem and periphery[35]. We find that, among the different RN regions, Ox1R is dominantly expressed in DRD while Ox2R is dominantly expressed in DRV, and only Ox1R could be clearly detected in serotonergic neurons in MRV and RPa. In line with this, a previously published RNA-seq data study shows that Ox2R is dominantly expressed in serotonergic neurons in DR, less expressed in the relatively dorsal and median regions of MR, and rarely expressed in ventral MR or caudal RN[20]. In addition, a recently published scRNA-seq data study shows that Ox1R is expressed at different levels among 5 subtypes of serotonergic neurons in DR[30].

Both Ox1R and Ox2R mediate an excitatory direct response upon orexin stimulation in serotonergic neurons, as found in our study and previous studies by others[33,36,37]. Consistent with the

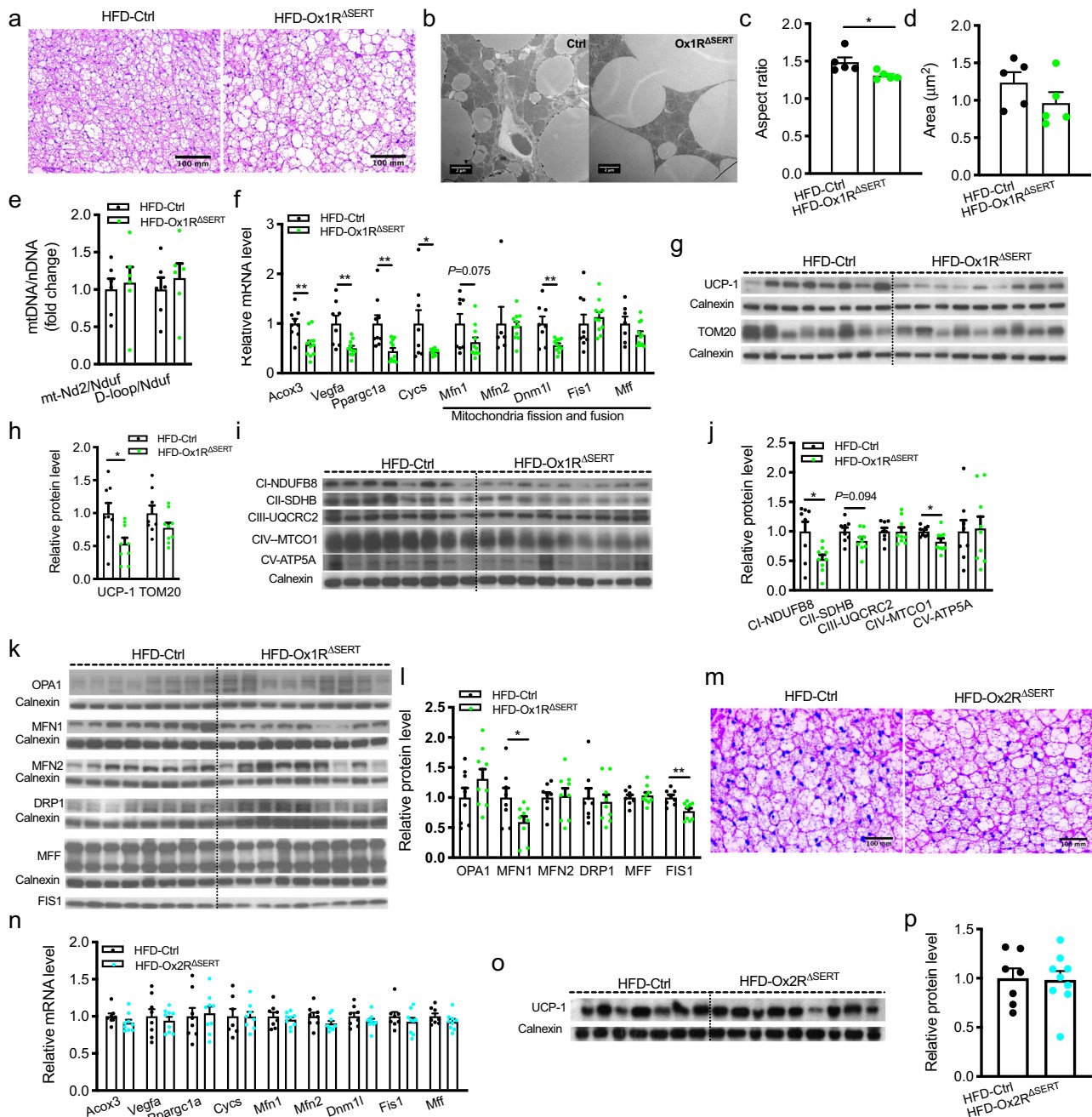

**Fig. 7 Increased fat and impaired mitochondrial function in brown adipose tissue (BAT) of Ox1R$^{\Delta SERT}$ mice while BAT morphology and mitochondrial function are unaltered in Ox2R$^{\Delta SERT}$ mice fed a high-fat diet (HFD). a** H&E staining of BAT of control or Ox1R$^{\Delta SERT}$ mice. HFD-Ctrl, $n = 8$; HFD-Ox1R$^{\Delta SERT}$, $n = 10$. **b** Representative electronic microscope (EM) images of BAT of control and Ox1R$^{\Delta SERT}$ mice, and quantification of **c** mitochondrial area and **d** mitochondrial aspect ratio. $n = 5$. $p = 0.033$. **e** Relative mitochondrial DNA content of control and Ox1R$^{\Delta SERT}$ mice. $n = 6$. **f** Gene expression levels in BAT of control and Ox1R$^{\Delta SERT}$ mice. Control: $n = 6$ (*Mfn2*), 7 (*Mff*), 8 (*Vegfa, Cycs, Mfn1, Dnm1l*) and 9 (*Acox3, Ppargc1a*, Fis1); Ox1R$^{\Delta SERT}$, $n = 11$. $p = 0.0026$ (*Acox3*), 0.0031 (*Vegfa*), 0.0038 (*Ppargc1a*), 0.024 (*Cycs*), 0.075 (*Mfn1*) and 0.0035 (*Dnm1l*). **g, h** Western blot images and quantification of UCP-1 and TOM20 protein, **i, j** OXPHOS protein, and **k, l** mitochondrial fusion and fission protein in BAT of control and Ox1R$^{\Delta SERT}$ mice. UCP1 and FIS1 were from the same western blot gel and thus shared the loading control. HFD-Ctrl, $n = 8$; HFD-Ox1R$^{\Delta SERT}$, $n = 9$. $p = 0.020$ (UCP−1), 0.015 (CI-NDUFB8), 0.094 (CII-SDHB), 0.021 (CIV-MTCO1), 0.038 (MFN1) and 0.0039 (FIS1). **m** H&E staining of BAT of control and Ox2R$^{\Delta SERT}$ mice. HFD-Ctrl, $n = 11$; HFD-Ox2R$^{\Delta SERT}$, $n = 9$. **n** Gene expression levels in BAT of control and Ox2R$^{\Delta SERT}$ mice. HFD-Ctrl, $n = 8$ except for Cycs ($n = 7$); HFD-Ox2R$^{\Delta SERT}$, $n = 9$ except for Cycs ($n = 8$). **o, p** Western blot images and quantification of UCP-1 protein in BAT of control and Ox2R$^{\Delta SERT}$ mice. HFD-Ctrl, $n = 7$; HFD-Ox2R$^{\Delta SERT}$, $n = 9$. Scale bar: 100 mm in (**a, m**) and 2 μm (**b**). Data are represented as means ± SEM. *$p < 0.05$, **$p < 0.01$; as determined by unpaired two-tailed Student's $t$-test. Source data are provided as a Source Data file.

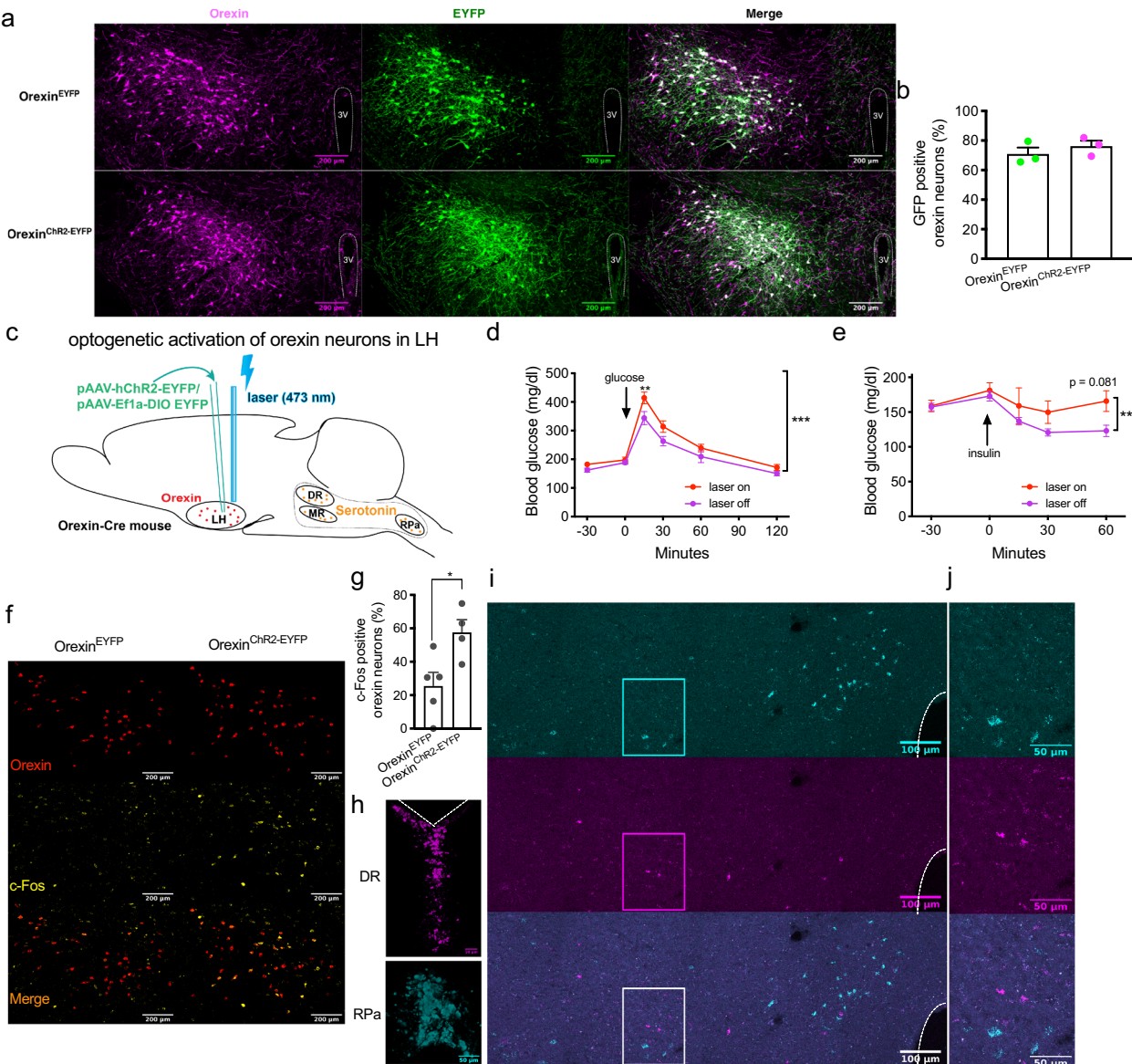

**Fig. 8 Optogenetic stimulation of orexin neurons impairs glucose tolerance. a** Representative images of immunostaining of orexin and EYFP in the lateral hypothalamus (LH) of Orexin-Cre mice injected with Cre-dependent adeno-associated virus pAAV-EYFP (Orexin^EYFP) or pAAV-ChR2-EYFP (Orexin^ChR2-EYFP). $n = 3$. **b** Quantification of percentages of GFP positive neurons in orexin neurons of mice injected with AAV. $n = 3$. Magenta, orexin; green, EYFP. **c** Schematic drawing of the strategy of optogenetic stimulating of orexin neurons. **d** Glucose tolerance test and **e** insulin tolerance test of Orexin^ChR2-EYFP mice with (laser on) or without (laser off) laser illumination in LH. **d** $n = 10$, $p = 0.0056$ (15 min); **e** $n = 9$, $p = 0.081$ (60 min). **f** Representative images of RNAscope in situ hybridizations of c-Fos and orexin in LH and **g** the quantification of percentages of c-Fos positive neurons in orexin neurons after laser illumination. $n = 5$ or 4. $p = 0.027$. Red, orexin; yellow, c-Fos. **h** Representative images of retrobeads injected in dorsal raphe nucleus (red beads) and raphe pallidus (green beads). **i** Representative images of retrobeads traveled to LH, in which the squares-indicated regions were amplified in (**j**). $n = 5$. Magenta, red beads; cyan, green beads. Scale bar: 200 μm (**a**, **f**), 100 μm (**i**) or 50 μm (**h**, **j**). Data are represented as means ± SEM. *$p < 0.05$, **$p < 0.01$, ***$p < 0.001$; as determined by unpaired two-tailed Student's *t*-test (**g**) or two-way ANOVA followed by Sidak's post hoc test (**d**, **e**). Two-way ANOVA revealed a significant main effect of genotype in (**d**) ($F (1, 108) = 15.75$, $p = 0.0001$), and (**e**) ($F (1, 80) = 7.00$, $p = 0.0098$). Source data are provided as a Source Data file.

differential expression pattern of orexin receptors in DR, $Ca^{2+}$ imaging reveals that, in DRV, the unselective endogenous agonist orexin A-induced excitation of serotonergic neurons is mainly mediated by Ox2R. Both electrophysiological and GCaMP experiments find that the excitation of serotonergic neurons by orexin B, which is more selective for Ox2R activation[38], is attenuated by Ox2R deletion in serotonergic neurons but not by Ox1R deletion. Together with our RNAscope findings for Ox1R or Ox2R in serotonergic neurons of Ox1R^ΔSERT or Ox1R^ΔSERT mice, respectively, this suggests that our Cre-loxP

recombination-mediated conditional knockout model is specific and efficient.

Deletion of either orexin receptor in serotonergic cells is insufficient to alter glucose metabolism and/or energy homeostasis in lean mice. However, it has opposing effects under conditions of HFD-induced obesity when the orexin system is more activated. This indicates that orexin signaling in serotonergic neurons is more crucial for maintaining energy homeostasis under obese conditions, compared to lean conditions. Consistent with this hypothesis, we found orexin neurons

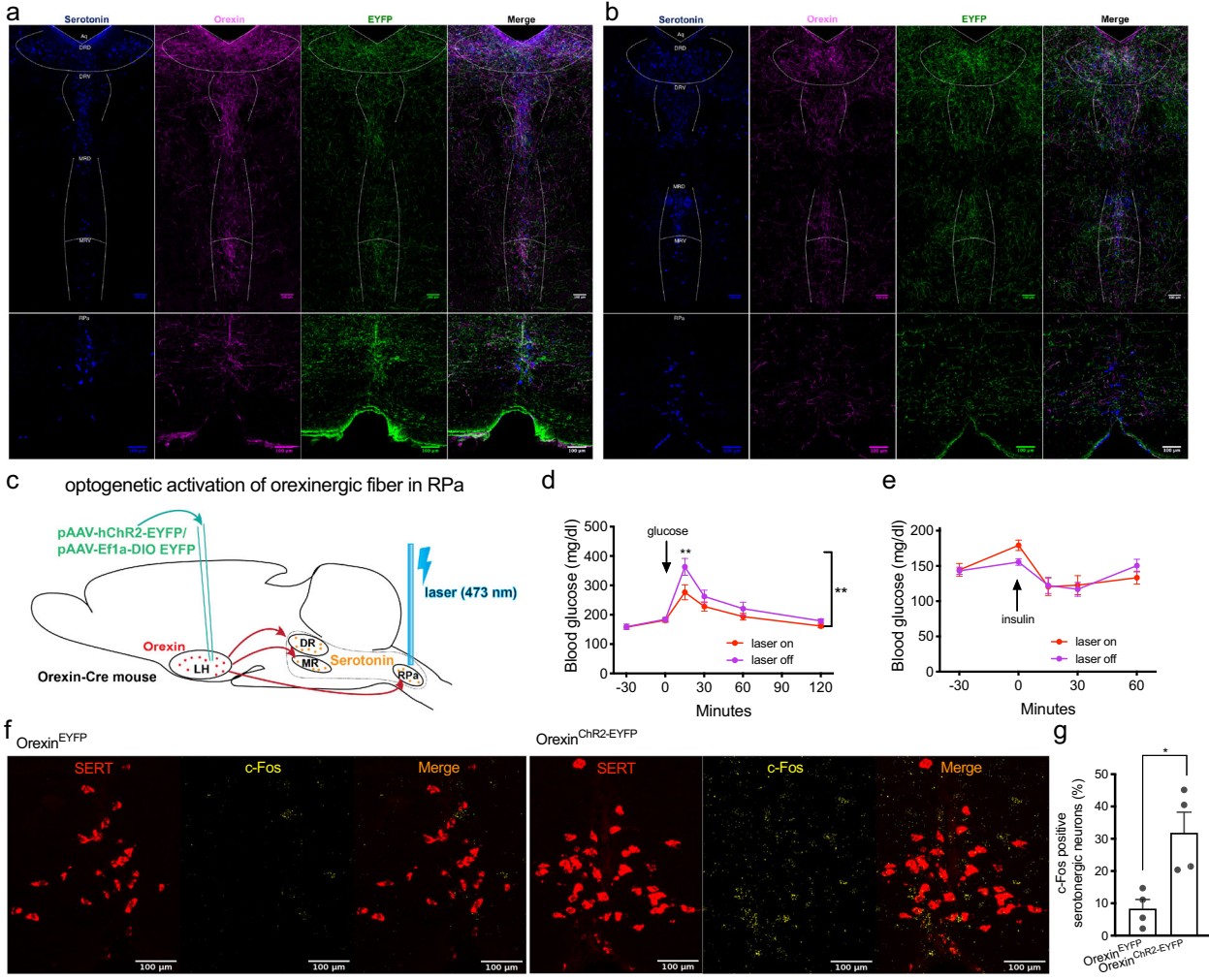

**Fig. 9 Optogenetic stimulation of orexinergic fibers in raphe pallidus (RPa) improves glucose tolerance. a** Representative images of immunostaining of orexin and EYFP in DR/MR (upper panel) and RPa (lower panel) of Orexin-Cre mice injected with Cre-dependent adeno-associated virus pAAV-EYFP or **b** pAAV-ChR2-EYFP virus (Orexin^ChR2-EYFP). $n = 3$. Blue, serotonin; magenta, orexin; green, EYFP. **c** Schematic drawing of the strategy of optogenetic stimulation of orexinergic fibers in RPa. **d** Glucose tolerance test and **e** insulin tolerance test of Orexin^ChR2-EYFP mice with (laser on) or without (laser off) laser illumination in RPa. $n = 8$. **d** $p = 0.0015$ (15 min). **f** Representative images of RNAscope in situ hybridization of c-Fos and serotonin transporter (SERT) in raphe pallidus and **g** the quantification of percentages of c-Fos positive neurons in serotonergic neurons after laser illumination. $n = 4$. $p = 0.015$. Red, SERT; yellow, c-Fos. Scale bar: 100 μm. Data are represented as means ± SEM. *$p < 0.05$, **$p < 0.01$; as determined by unpaired two-tailed Student's $t$-test (**g**) or two-way ANOVA followed by Sidak's post hoc test (**d**). Two-way ANOVA revealed a significant main effect of genotype in (**d**) ($F_{(1, 84)} = 9.34$, $p = 0.0030$). Source data are provided as a Source Data file.

were more activated in HFD-fed mice compared to CD-fed mice under fasting conditions. Previous reports also show that orexin receptors regulate energy metabolism differently in lean and obese mice. For example, overexpression of orexin decreases BW of mice fed a low-fat-diet or HFD, but deletion of Ox2R only abolishes the effect in mice on HFD[10].

Ox1R signaling in serotonergic cells protects against HFD-induced insulin resistance, due to changes in peripheral glucose uptake but not hepatic gluconeogenesis. Specifically, Ox1R signaling in serotonergic neurons activates BAT thermogenesis and glucose uptake. Ox1R deletion in serotonergic cells of obese mice resulted in larger lipid droplets in BAT and increased BAT weight. BAT whitening attracts immune cells and tissue chronic inflammation, which further increases insulin resistance, under obese conditions[39–42]. Mitochondria are in an equilibrium of fission and fusion, and the mitochondrial elongation/fusion facilitates oxidative phosphorylation that produces ATP from glucose[43–45]. We find impaired mitochondrial function, including markers of reduced mitochondrial fusion, oxidative

phosphorylation, lipid metabolism, and thermogenesis in BAT of obese mice that lack Ox1R in serotonergic neurons, suggesting that the capability of BAT to utilize glucose and fatty acids for thermogenesis and ATP production are both reduced. In line with this, BAT glucose uptake under steady-state conditions in hyperinsulinemic-euglycemic clamp studies is decreased in obese Ox1R^ΔSERT mice. Previous studies reported that BAT thermogenesis is activated by both the serotonin and the orexin systems. Orexin knockout mice display impaired thermogenesis due to the inability of brown preadipocytes to differentiate[46]. Another study fails to find detectable abnormalities in BAT development of orexin-deficient mice, and predicts that orexin controls BAT thermogenesis via a central pathway, based on the data that Ox1R is expressed low and Ox2R is undetectable in BAT[15].

Serotonergic neurons in the rostral RPa play a key role in the control of BAT thermogenesis[26]. We find abundant orexinergic fibers innervating RPa, and optogenetic activation of these orexinergic fibers improves glucose tolerance. Ox1R but not Ox2R is expressed in serotonergic neurons in RPa, and consistent

with this, BAT morphology and thermogenic factors remained unchanged when Ox2R is specifically deleted in serotonergic neurons. Therefore, our data suggest that Ox1R in serotonergic neurons in RPa is crucial to mediate the activation of BAT thermogenesis by orexin and serotonin systems, and thus to improve glucose metabolism.

Glucose uptake in skeletal muscle is also reduced in serotonergic-specific Ox1R-deleted obese mice. A previous study reported that orexin injection in the ventromedial hypothalamus stimulates skeletal muscle glucose uptake via activation of the sympathetic nervous system[47]. Indeed, orexin and serotonin systems both regulate the autonomic system[48,49]. However, further investigations are clearly needed to understand the mechanisms of Ox1R signaling in serotonergic neurons in the stimulation of skeletal muscle glucose uptake.

In contrast to Ox1R, Ox2R signaling in serotonergic neurons impairs glucose tolerance and contributes to insulin resistance induced by HFD feeding. The dominant Ox2R signaling in DRV could be important in glucose intolerance and insulin resistance upon HFD feeding. The impairment of glucose tolerance and insulin sensitivity by Ox2R in serotonergic neurons is mainly due to the impaired insulin-stimulated suppression of gluconeogenesis in liver. Although orexin deleted mice exhibit late-onset obesity, acute i.c.v. injection of orexin increases blood glucose levels[14]. In line with this, we find that optogenetic activation of orexin neurons in LH induces significant impairment of glucose tolerance and a trend toward impairment of insulin sensitivity by GTT and ITT. Central application of orexin increases the expression of G6Pase and Pepck in liver and the increase of Pepck is abolished by the deficiency of Ox2R but not Ox1R, suggesting that Ox2R mainly mediates the orexin-induced increase of gluconeogenesis in liver[14]. This supports our hypothesis that Ox2R signaling in serotonergic neurons, especially in DRV, potentially plays an important role in orexin-induced glucose elevation via increasing liver gluconeogenesis.

The orexin system shows a circadian rhythm and is also changed according to the nutrition state, which is related to its bidirectional regulation of blood glucose[12,14]. The differential roles of Ox1R and Ox2R in serotonergic neurons could partially explain the underlying mechanisms of this bidirectional regulation. Obese Ox1R$^{\Delta SERT}$ and Ox2R$^{\Delta SERT}$ mice exhibit opposite changes in GIR in euglycemic-hyperinsulinemic clamp experiments, for which mice are fasted for 4 h in the morning. Obese Ox1R$^{\Delta SERT}$ mice show impaired glucose tolerance in GTT after a 6-h (in the morning) but not 16-h (overnight) fasting period, while obese Ox2R$^{\Delta SERT}$ mice exhibit a more significant improvement in glucose tolerance in GTT after a 16-h fasting period compared to 6-h fasting. Ox1R and Ox2R signaling in serotonergic cells seems to dominantly regulate glucose metabolism under a shorter fasting condition at light phase or after a longer fasting time at dark phase, respectively. On the other hand, conditional knockout of orexin receptors in serotonergic neurons of obese mice significantly alters glucose tolerance during a GTT and alters glucose uptake and gene/protein expression levels in peripheral tissue and insulin sensitivity during euglycemic-hyperinsulinemic clamp experiments under fasting conditions, while insulin tolerance during ITT, BW and body composition under random-fed conditions remained unaltered. This indicates that orexin signaling in serotonergic neurons could be more relevant under nutrition-restricted conditions with respect to the control of glucose metabolism. Moreover, in random-fed obese Ox1R$^{\Delta SERT}$ mice (21-wk old), impaired BAT function was detected at molecular and morphological levels. These clear changes in BAT could be late-onset and the time of HFD exposure may not have been sufficient to translate these effects into significant changes in BW or body composition, while compensation of other peripheral organs is

also possible. Clearly, further well-designed studies are needed to investigate the underlying mechanisms of the above findings.

Orexin acutely promotes feeding but in long term also promotes energy expenditure, in part by regulating spontaneous physical activity and supporting a sympathetic tone, which dominates in the control of energy balance. The serotonin system is also well known to be involved in the control of food intake and body weight[24,25]. However, in our study, deletion of orexin receptors in serotonergic neurons fails to alter locomotor activity or feeding behavior under random-fed conditions. This suggests that the regulatory role of orexin signaling in serotonergic neurons in glucose metabolism and energy homeostasis is independent of feeding or spontaneous physical activity.

We found intermingled and distinct LH cells projecting to DR or RPa. The boundaries of the heterogeneous RN subregions are not clear and their projections overlap, so it is important to define serotonergic neuron subtypes by both molecular and spatial specificity. Most strategies in previous reports about the differential functions of subregions of RN used micro-injections of drugs targeting the serotonergic system or analyzed serotonergic neuronal activity in experimental disease models[50]. The specific roles of subtypes of DR serotonergic neurons in energy homeostasis have been rarely studied so far. A recent study using viral-genetic methods shows that the subcortical-projecting serotonergic neurons preferentially localize in DRD and the cortical-projecting ones localized more in DRV. Further, they could show that the subcortical amygdala- and prefrontal-cortex-projecting DR neurons exhibit opposite responses to aversive stimuli and differentially regulate behaviors upon exposure to stress[51]. Our analysis reveals that the most differently expressed gene between Ox1R- and Ox2R-dominant serotonergic neurons is VGLUT3, and suggests more Ox2R-dominant serotonergic neurons are glutamatergic than Ox1R-dominant neurons. VGLUT3 has been shown to be higher expressed in serotonergic neurons in DRV compared to those in DRD, and it increases serotonin-1A receptor (5HT-1A)-mediated neurotransmission in RN and accelerates serotonin release at a specific subset of serotonergic nerve terminals[51,52]. Furthermore, DR neurons expressing vesicular transporters for gamma-aminobutyric acid (GABA), Vgat (Slc32a1), or for glutamate, VGLUT3, were shown to increase or suppress food intake, respectively[53]. According to scRNA-seq data, GABA receptors (Gabbr1 and Gabrg3) and VGLUT3 are enriched in different subtypes of serotonergic neurons, potentially suggesting that different subtypes of serotonergic neurons modulate food intake via distinct mechanisms[30,53]. Ox1R and Ox2R signaling relate to both the spatial and neurochemical heterogeneity of serotonergic neurons in DR. Together with our data, these reports support the hypothesis that serotonergic neurons in DRD and DRV have distinct roles in the modulation of glucose metabolism and energy homeostasis.

Only Ox1R but not Ox2R are expressed in serotonergic neurons in MRV. Interestingly, Ox2R is abundantly expressed in non-Sert-expressing neurons in MRV, which implicates the possibility of their crosstalk with Ox1R-expressing serotonergic neurons in the same area. Serotonergic neurons in DR and MR reciprocally connect to neurons in the hypothalamus (HTN), implying a functional role in regulating energy metabolism. The projections of DR and MR were mainly studied with neuron-subtype-unspecific tracing methods, and are considered more diverse than their inputs[54,55]. Distinct roles for MR and DR serotonergic neurons have been described and some researchers even suggest antagonism between them[56,57]. It will be interesting to elucidate the functional roles of orexin receptor signaling in this region in future studies.

Collectively, our study reveals differential expression patterns and functional roles of Ox1R and Ox2R signaling in serotonergic

neurons in the control of peripheral glucose metabolism, BAT thermogenesis, and liver gluconeogenesis. Further detailed studies to understand this complex neuronal network and underlying cellular mechanisms will help to develop possible therapies targeting the orexin or serotonin systems for obesity.

## Methods

**Animal care**. All animal procedures were conducted in compliance with protocols approved by the local government authorities (Bezirksregierung Cologne, Germany) and were in accordance with National Institutes of Health guidelines. Permission to maintain and breed mice as well as all for experimental protocols in this study was issued by the Department for Environment and Consumer Protection - Veterinary Section, Cologne, North Rhine-Westphalia, Germany (84-02.04.2015.A335). Mice were housed in groups of 3–5 at 22 °C–24 °C using a 12-h light/12-h dark cycle, with humidity of 50–70%. Animals had ad libitum access to water and food at all times, and food was only withdrawn if required for an experiment. Animals were fed a NCD (ssniff[R] R/M-H Phytoestrogenarm), which contains 57 KJ% calories from carbohydrates, 34 KJ% calories from protein, and 9 KJ% calories from fat, a CD (ssniff[R] EF D12450B * mod. LS), which contains 67 KJ% calories from carbohydrates, 20 KJ% calories from protein and 13 KJ% calories from fat, or a HFD (ssniff[R] EF acc. D12492 (I) mod.), which contains 21 KJ% calories from carbohydrates, 19 KJ% calories from protein and 60 KJ% calories from fat.

BL/6 mice were purchased from Charles River, France. tdTomato fl/fl mice were purchased from The Jackson Laboratory (B6;129 S6-Gt(ROSA)26Sor[tm9(CA6-tdTomato)Hze], stock No: 007905), and have been previously described[29]. Slc6a4-Cre (Sert-Cre) mice were imported from the Mutant Mouse Resource & Research Center supported by NIH (stock number: 017260-UCD)[58]. ChR2-tdTomato fl/fl mice were obtained from The Jackson laboratory (B6; 129S-Gt(ROSA) 26Sortm32(CAG-COP4∗ChR2(H134R)tdTomato, Ai27, stock No: 012567)[59].

**Generation of Ox1R fl/fl mice and Ox2R fl/fl mice**. Ox1R flox (Ox1R fl/fl, Supplementary Fig. 2a–d) and Ox2R flox (Ox2R fl/fl, Supplementary Fig. 2e–h) mice were generated in our facility.

To create a conditional Ox1R allele, a targeting vector was constructed which flanks exon 4, 5, and 6 of the Ox1R gene (Hcrt1) by loxP sites by inserting the two homology arms and the loxP-flanked gene segment into the GK12TK vector using standard cloning techniques. The 4.3 kb short arm of homology was amplified using primers Ox5SA (5′-GCGGCCGCTCAGCACGACATGCTCAGAGA-3′) and Ox3SA (5′-GCGATCGCATTATCTGCACTGCGAATATAGC-3′). Primers Flox5 (5′-GGCGCGCCTATGTTCCAATGTCAGGGACC-3′) and Flox3 (5′-GGCCGGCCACCCATTCTTGCTGGTGAGGT-3′) were used in a PCR reaction to amplify the 2.2 kb loxP-flanked gene segment. The 5 kb long arm of homology was amplified by PCR using primers Ox5LA (5′-AAGCTTATGAAGGGAAGGCCCAGGACT-3′) and Ox3LA (5′-TTAATTAACCTCTGTTTCCTGACTTAGAG-3′). C57/BL6-derived Bruce-4 embryonic stem cells were transfected with the linearized targeting vector and subsequently selected for G418 and GANC resistance. Four hundred single clones were isolated from which three were shown to contain the external loxP-site using probe B that was amplified by PCR using primers S5C (5′-AGT TGTGAGCATGTGCAAGG-3′) and S3C (5′-CTCTAAACATCACATATC-3′). Single integration of the targeting vector was confirmed by using a probe in Southern Blot analysis against the neomycin resistance gene at HindIII digested genomic clonal DNA resulting in a single 11.6 kb band. Positive clones were injected into CB20 blastocysts to obtain chimeric mice (Ox1R[FL-neo]), which were further intercrossed with Flp-deleter mice[60] to achieve both, germline transmission and removal of the FRT flanked neo resistance cassette.

To create a conditional Ox2R allele, a targeting vector was constructed which flanks exon 2 of the Ox2R gene (Hcrt2) by loxP sites by inserting the two homology arms and the loxP-flanked gene segment into the GK12TK vector using standard cloning techniques. The 1.6 kb short arm of homology was amplified using primers Ox5SA (5′-GCGGCCGCAGACAAGCCTCTGGGCAAAGT-3′) and Ox3SA (5′-CC GCGGCTTAATCTTAGC CTTGGGGAGG-3′). Primers Flox5 (5′-GGCGCGCCA CACATGCTGCTATACCTAT-3′) and Flox3 (5′-GGCCGGCCTCATTAGTTTG TTCAGATCATCTC-3′) were used in a PCR reaction to amplify the 0,8 kb loxP-flanked gene segment. The 5 kb long arm of homology was amplified by PCR using primers Ox5LA (5′-CTTAAGTTAAGACATCCCTAGCTCAAA-3′) and Ox3LA (5′-TTAATTAACACCTCCAAAGGACCTGAATT-3′). C57/BL6-derived Bruce-4 embryonic stem cells were transfected with the linearized targeting vector and subsequently selected for G418 and GANC resistance. Single clones were isolated from which 2 were shown to contain the external loxP-site using probe B that was amplified by PCR using primers S5C (5′-AACCCATACCCTGACCCTTC-3′) and S3C (5′-TTTCCAAAATGCCTTTCCAG-3′). Single integration of the targeting vector was confirmed by using a probe in Southern Blot analysis against the neomycin resistance gene at HindIII digested genomic clonal DNA resulting in a single 12.7 kb band. Positive clones were injected into CB20 blastocysts to obtain chimeric mice (Ox2R[FL-neo]), which were further intercrossed with Flp-deleter mice[60] to achieve both, germline transmission and removal of the FRT flanked neo resistance cassette.

**Generation of Orexin-IRES-Cre mice**. Orexin-IRES-Cre mice were generated, validated, and kindly provided by Drs. D.K., T.E.S., and B.B.L. Briefly, a BAC clone containing the mouse Ore/Hcrt genomic sequence was used to generate the targeting construct. A PCR amplicon containing the IRES-Cre-FRT-NEO-FRT cassette flanked by 70 bp of homologous sequence at both ends, which matched the 3′ end UTR site of Orx/Hcrt allele, was constructed and transformed into electrocompetent DY380 cells that had been previously transformed with the aforementioned Ore/Hcrt BAC[61], followed by temperature-induced recombineering. A targeting construct was derived from this modified BAC spanning the region 5 kb upstream of the IRES site and 2 kb downstream of the Cre site inserted into 3′-end genomic region. The resulting targeting construct was then electroporated into mouse W4/129S6 embryonic stem (ES) cells and injected into blastocysts to generate chimeras. Male chimeras were bred to mice bearing a flp-recombinase transgene to remove the neomycin selection marker and to Ai14 cre reporter line (Jax 007908) to verify cre activity in the hypothalamus. Detailed description and characterization of this mouse line will be reported separately (Howard et al. unpublished).

**GTTs and ITTs**. GTTs were performed on 12- or 15-week-old male animals that had been fasted for 6 h or overnight for 16 h from 8:30 am or 6 pm, respectively. ITTs were performed on 11-week-old random-fed male mice around 9 am. Animals received an intraperitoneal injection of 20% glucose (10 ml/kg body weight; KabiPac) or insulin (0.75 U/kg body weight; Sanofi-Aventis) into the peritoneal cavity, respectively. Glucose levels were determined in blood collected from the tail tip using an automatic glucose monitor (Contour[R], Bayer), immediately before and 15, 30, and 60 min after the injection, with an additional value determined after 120 min for the GTTs.

**Relative mitochondrial DNA quantification**. BAT was dissected from 21-week-old male HFD-fed mice. Total DNA from BAT was isolated and the DNA level was detected with Power SYBR[R] Green PCR Master Mix (ThermoFisher Scientific). Primers for mitochondrial Nd2 and nuclear Nduf were from the publication by Jais et al.[62], and primers for mitochondrial D-loop region (fwd: 5′-GGTTCTTACT TCAGGGCCATCA-3, rev: 5′-GATTAGACCCGATACCATCGAGAT-3′) were designed with Primer Express software (Applied Biosystems). The relative mitochondrial DNA content to the nuclear DNA was determined with $2^{-ddCT}$ method.

**Analysis of gene expression**. BAT was dissected from 21-week-old male HFD-fed mice. Total RNA from BAT was isolated using the RNeasy[R] Lipid Tissue Mini Kit (Qiagen). cDNA was prepared using the High-Capacity cDNA Reverse Transcription Kit (Applied Biosystems). mRNA levels were then determined with real-time quantitative RT-PCR using TaqMan[R] Gene Expression Master Mix (ThermoFisher Scientific) and the respective probes, according to the manufacturer's instructions. Gapdh was used as the internal control. Relative expression was determined using a comparative method ($2^{-ddCT}$). Probes purchased from ThermoFisher Scientific are shown in Supplementary Table 1.

**RNA sequencing of BAT**. BAT was dissected at the end of hyperinsulinemic-euglycemic clamp experiments. The tissue was frozen at −80 °C until the RNA was extracted with DNase digestion using the RNeasy[R] Lipid Tissue Mini Kit (Qiagen) and RNase-Free DNase Set (Qiagen), following the user's manual. RNA integrity was detected with the Agilent RNA 6000 Nano Kit (5067-1511, Agilent Technologies) and the Agilent 2100 Bioanalyzer. RNA libraries were prepared using the Illumina® TruSeq® mRNA stranded sample preparation Kit. Library preparation started with 1 µg total RNA. After poly-A selection (using poly-T oligo-attached magnetic beads), mRNA was purified and fragmented using divalent cations under elevated temperatures. The RNA fragments underwent reverse transcription using random primers. This is followed by second-strand cDNA synthesis with DNA Polymerase I and RNase H. After end repair and A-tailing, indexing adapters were ligated. The products were then purified and amplified (14 PCR cycles) to create the final cDNA libraries. After library validation and quantification (Agilent tape station), equimolar amounts of the library were pooled. The pool was quantified by using the Peqlab KAPA Library Quantification Kit and the Applied Biosystems 7900HT Sequence Detection System. The pool was sequenced for 35 million reads using an Illumina NovaSeq6000 instrument and a PE100 sequencing protocol.

For the statistical analysis, we applied the community-curated nfcore rnaseq analysis pipeline version 1.4[63]. The gene-level quantification was carried out using Salmon 0.14.1[64] using the reference genome GRCm38 (https://www.ncbi.nlm.nih.gov/assembly/GCF_000001635.20). The differential gene expression analysis was done using the DESeq2 1.26.0[65] R package and yielded 265 genes (Fig. 6e). A gene-ontology term analysis of the 265 differentially expressed genes was carried out using the clusterProfiler 3.14.3[66] R package and yielded 530 differentially expressed GO-terms of class biological process. Applying the simplified method of clusterprofiler, which removes similar gene ontologies, yielded subsequently a list of 131 differentially expressed biological process terms. Figure 6f shows the top 15 differentially expressed terms of that list.

**Western blotting**. BAT was dissected from 21-week-old male mice on HFD and homogenized in RIPA buffer (Sigma-Aldrich) added with cOmplete[TM] Protease

Inhibitor Cocktail (Roche), using a FastPrep instrument (MP Biomedicals) and bulk beads (Bertin Corp.). The protein samples were separated on 10% Criterion$^{TM}$ TGX$^{TM}$ Precast Midi Protein Gel (BioRad) and transferred with Trans-Blot$^R$ Turbo$^{TM}$ Midi PVDF Transfer pack (BioRad). After blocking with 5% non-fat milk or a blocking reagent (Roche) in Tris-buffered saline containing 0.1% Tween-20 (TBST) at room temperature (RT) for 1 h, the membranes were incubated overnight at 4 °C with antibodies raised against pSer$^{473}$-Akt (1:1000, #4060, rabbit mAb, Cell Signaling Technology), Akt (1:1000, #4685, rabbit mAb, Cell signaling Technology), G6Pase (1:250, sc-25840, rabbit pAb, Santa Cruz Biotechnology), UCP-1 (1:200, sc-6528, goat pAb, Santa Cruz Biotechnology), Tom20 (1:100, sc-17764, mouse mAb, Santa Cruz Biotechnology), OPA1 (1:1000, 612607, mouse mAb, BD Biosciences), MFN1 (1:1000, ab57602, mouse mAb, Abcam), MFN2 (1:1000, ab56889, mouse mAb, Abcam), DRP1 (1:1000, #8570, rabbit mAb, Cell Signaling Technology), MFF (1:2000, 17090-1-AP, rabbit pAb, Proteintech$^R$), FIS1 (1:500, 10956-1-AP, rabbit pAb, Proteintech$^R$) or complex I-V subunits in the respiratory chain (1:1000, Total OXPHOS Rodent WB Antibody Cocktail, containing 5 mouse mAbs, ab110413, Abcam). Calnexin (1:5000, 208,880, rabbit pAb, Calbiochem$^R$) was used as the loading control. The secondary antibodies (1:3000) were then incubated at RT for 1 h. Peroxidase conjugate goat anti-rabbit IgG (A6154), goat anti-mouse IgG (A4416) and mouse anti-goat IgG (A9452) were purchased from Sigma-Aldrich. The signal was detected with SuperSignal™ West Dura Extended Duration Substrate (ThermoFisher Scientific), using films or camera (Vilber Smart Imaging). Films were developed with X-ray film processor Curix 60 (AGFA Healthcare). When necessary, membranes were stripped in stripping buffer (62.5 mM Tris pH 6.8, 2% SDS, 0.7% β-mercaptoethanol; 56 °C, 30 min), washed twice in TBST, blocked in blocking buffer, and reprobed with the respective primary antibodies. Band densities were analyzed with Image J/FIJI (version 1.50d, http://imagej.net). Data were normalized to protein expression levels in the control group.

**Imaging**. BAT was post-fixed after sacrificing. Brains were dissected from perfused mice, except for optogenetic experiments. Anesthetized mice were perfused with 0.9% saline, followed by 4% paraformaldehyde (PFA) in 0.1 M phosphate-buffered saline (PBS). Brains were removed, post-fixed in 4% PFA for the indicated time, and dehydrated in 20% sucrose in 0.1 M PBS overnight. After embedding in Leica Tissue Freezing medium, brains were stored at −80 °C until cutting. For optogenetics, head was cut and directly put in 4% PFA at 4 °C for 7 days until brain dissection, and then brains were dehydrated and processed as above. Each brain/mouse represents one biological replicate, and at least two images, as technical replicates, were taken for each brain/mouse.

*Haematoxylin and eosin (H&E) staining of BAT*. BAT was dissected from 21-week-old male mice on a HFD. It was post-fixed in 4% PFA at 4 °C for more than 16 h and embedded with pre-set program in Shandon$^{TM}$ Excelsior$^{TM}$ ES Tissue Processor Accessories (ThermoFisher Scientific). In general, tissue was put in formalin for 1 h twice, 70% ethanol for 2 h twice, 96% ethanol for 2 h twice, 100% ethanol for 1 or 2 h three times, xylene for 1 or 2 h three times, and liquid wax for 1 h to 1.67 h three times, at 45 °C. Tissues were then embedded in blocks using Leica EG1150 H Heated Paraffin Embedding Module and Leica EG1150 C Cold Plate for Modular Tissue Embedding System. Sample blocks were stored at RT until cutting. Slices (5 μm) were cut with Leica RM2255 Fully Automated Rotary Microtome. Slices were deparaffinized with xylene for 20 min, put into isopropanol for 2 min, gradually hydrated with diluted ethanol and water, stained with Mayer´s haematoxylin solution (Sigma-Aldrich) followed by eosin solution (Sigma-Aldrich) after washing, washed in water, gradually dehydrated in diluted ethanol, put in xylene and finally mounted with Cytoseal XYL (ThermoFisher Scientific). Slides were stored at RT, and imaged with Zeiss Imager M2 microscope and the software Zen 2 (Carl Zeiss AG).

*Electronic microscope images of BAT*. BAT samples were isolated from HFD-fed male mice (12–15 weeks) and fixed in 2% glutaraldehyde and 2% formaldehyde in 0.1 M cacodylate buffer (pH 7.2) for 48 h at 4 °C. Samples were rinsed in 0.1 M cacodylate buffer (pH 7.2), post-fixed with 1% OsO4 in 0.1 M cacodylate buffer (pH 7.2) for 3 h at 4 °C, dehydrated through an ethanol series, transferred to propylene oxide, and embedded in epoxy resin. Semi-thin sections of 500 nm were cut, followed by cutting into ultrathin sections of 70 nm, with a diamond knife (Diatome, Biel, Switzerland) on an ultramicrotome (EM-UC6, Leica). Ultrathin sections were place on a 100 mesh grid, contrasted with 1.5% uranyl acetate for 15 min at 37 °C, put in lead citrate for 4 min after washing, and dried after additional washing. Images were taken on a transmission electron microscope (Zeiss EM109 and JEOL JEM 2100Plus) at RT. Mitochondrial size and aspect ratio were analyzed with Image J/FIJI (version 1.50d).

*Immunostaining in lateral hypothalamus and raphe nuclei*. Brains were post-fixed for 6 h at 4 °C, and cut (30 μm) with Leica CM3050 S Research Cryostat. Sections were incubated in 0.3% glycine for 10 min after washing in 0.1 M PBS for 2 × 10 min. After washing in PBS for another 10 min, sections were incubated in 0.03% SDS (in PBS) for 10 min before they got blocked with 3% donkey serum (in PBS, 0.25% Triton X-100) for 1 h at RT. Afterwards, they were incubated with primary antibodies, including goat anti-orexin A (1:250, sc-8070, goat pAb, Santa

Cruz Biotechnology), chicken anti-GFP (1:1000, ab13970, chicken pAb, Abcam), rabbit anti-TPH2 (1:1000, #51124, rabbit mAb, Cell Signaling Technology) and/or rabbit anti-serotonin (1:400, S5545, rabbit pAb, Sigma-Aldrich), for overnight at RT, washed in PBS for 3 × 10 min, and incubated with secondary antibodies (1:500), including Alexa Fluro 594 donkey anti-goat, FITC donkey anti-chicken, Alexa Fluro 488 donkey anti-rabbit, Alexa Fluro 647 donkey anti-rabbit (a11058, sa 1-7200, a21206 and a31573, Invitrogen), for 1 h at RT. After washing in PBS for 3 × 10 min, slices were mounted and covered with VECTASHIELD Antifade Mounting Medium with DAPI (Vector Laboratories). Slices were stained together at one time for each experiment to have identical conditions for comparable signals. Slides were stored at 4 °C until imaging. Images were obtained with confocal laser scanning microscope Leica TCS SP8 and manually analyzed with Image J/FIJI (version 1.50d).

*RNAscope fluorescent in situ hybridization in lateral hypothalamus and raphe nuclei*. Brains from BL/6 mice (14 weeks, fasted for 6 h) on CD/HFD, and brains from Sert$^{tdTomato}$, Ox1R$^{ΔSERT/tdTomato}$, and Ox2R$^{ΔSERT/tdTomato}$ mice (13–18 weeks) on NCD were post-fixed for 20–22 h at RT and cut (20 μm). Slides were stored at −80 °C until staining. All reagents were purchased from Advanced Cell Diagnostics and the staining was performed with the RNAscope Multiplex Fluorescent v2 kit and RNAscope 4-Plex Ancillary Kit (ACD, Advanced Cell Diagnostics) according to the user manual. Probes for Ox1R (*Hcrtr1*, 471561-C3, ACD) and Ox2R (*Hcrtr2*, 471551, ACD) were designed according to the loxP-flanked region. The probes for tdTomato (*tdTomato*, 317041-C2, and C3), c-Fos (*c-Fos*, 316921-C4), TPH2 (*Tph2*, 318691), Pet1 (*Fev*, 413241-C2), SERT (*Slc6a4*, 315851-C2), VGLUT3 (*Slc17a8*, 431261-C2), and orexin (*Hcrt*, 490461-C2) were commercially available by ACD. In brief, slides were briefly washed in diethylpyrocarbonate (DEPC)-treated Millipore water, air dried and then dried at 60 °C overnight. On the second day, slides were treated with hydrogen peroxide (H$_2$O$_2$) for 10 min at RT, washed in water and boiled in Target Retrieval solution (around 99.4 °C) for 8–10 min. After a brief washing in water and dehydration in absolute ethanol, slides were incubated with protease IV for 30 min at RT. Slides were washed again in water and hybridized with the mixture of probes in different channels for 2 h in a humidified chamber at 40 °C. Afterwards, the hybridization was amplified with AMP 1 for 30 min, AMP 2 for 30 min, and AMP 3 for 15 min. The signal was then developed for each channel. For example, for channel 1, slides were incubated with HRP-C1 for 15 min, the fluorophore for 30 min, and HRP blocker for 15 min. All amplification and development were performed at 40 °C, and 2 × 2 min of washing in ACD wash buffer was performed after each step. Finally, the slides were counterstained with DAPI for 1 min, mounted with Pro-long$^{TM}$ Gold Antifade reagent with DAPI (Invitrogen), and covered with coverslips. For the RNAscope-immunostaining combined experiments, after the last washing step of RNAscope, slides were directly incubated with blocking buffer and processed with routine immunostaining methods.

Slides were dried and stored at 4 °C. Imaging was performed with confocal laser scanning microscope Leica TCS SP8. The raphe nuclei were automatically analyzed with the software Halo 2.0 (Indica Labs) in Fig. 1a–f and Fig. 2. Serotonergic neurons were identified according to the fluorescence intensity of tdTomato, and Ox1R and Ox2R signal in serotonergic neurons was detected and analyzed. Other images wer/e manually analyzed with Image J/FIJI (version 1.50d).

**Food intake and locomotor activity**. Male mice (20-week old) on NCD and HFD under random-fed conditions were analyzed. Food intake, locomotor activity, and indirect calorimetry measurements were made in a PhenoMaster System (TSE Systems) as previously described[67].

**Hyperinsulinemic-euglycemic clamp studies**. Surgical implantation of catheters in the jugular vein was performed in ~17-week-old HFD-fed male mice as previously described[68]. Mice recovered from surgery for ~1 week and only mice that regained at least 90% of pre-surgery bodyweight were included in further experiments. After fasting for 4 h, mice were placed into chambers, which allow them to move freely throughout the clamp experiment. After a bolus infusion (0.8 μCi) of D-[3-$^3$H]glucose (PerkinElmer) tracer solution, the tracer was infused continuously (0.04 μCi/min). After basal infusion, blood was collected for the determination of basal parameters. Then, insulin (Novo Nordisk) was infused at a fixed rate (6 μU/g/min) during the clamp experiment. Blood glucose levels were determined every 10 min (Glucose 201 RT System, HemoCue$^R$), and the infusion of 40% glucose (bela-pharm) was adjusted to maintain physiological blood glucose levels (120–160 mg/dl). Steady state was ascertained when glucose measurements were constant for at least 30 min at a fixed glucose infusion rate and was achieved within 100 to 120 min of the clamp experiment. At the end of the steady state, a bolus (10 μCi) of 2-deoxy-D-[1-$^{14}$C]-glucose (2DG; American Radiolabeled Chemicals, Inc.) was infused. Blood was collected 2, 7, 15, 25, and 35 min after the bolus. At the end of the experiment, mice were sacrificed and tissues were dissected. Plasma was stored at −20 °C, and liver, SM, WAT, and BAT were stored at −80 °C until further analysis.

Plasma [3-$^3$H] glucose radioactivity of basal and steady state was measured and glucose turnover rate (mg × kg$^{-1}$ × min$^{-1}$) was calculated as previously described[68]. Plasma 2-[1-$^{14}$C]-Deoxy-D-glucose radioactivity was directly measured in the liquid scintillation counter. WAT, BAT, and SM lysates were

processed through ion-exchange chromatography columns (Poly-Prep Prefilled Chromatography Columns, AG1-X8 formate resin, 200–400 mesh dry; Bio-Rad) to separate 2DG from 2DG-6- phosphate (2DG6P). In vivo glucose uptake for WAT, BAT, and skeletal muscle (nmol $\times$ g$^{-1}$ $\times$ min$^{-1}$) was calculated based on the accumulation of 2DG6P in the respective tissue and the disappearance rate of 2DG from plasma as described previously[69]. Serum human insulin concentrations at baseline levels and at the end of the clamp studies were measured with DRG® Ultra Sensitive Insulin ELISA Kit (EIA-2337).

**Virus injection and optical fiber implantation.** 13-week-old male Orexin-Cre mice received 1 mg/ml of tramadol in drinking water 2 days before the surgeries. On the day of surgery, mice were anesthetized with isoflurane and a bolus of buprenorphine (0.1 mg/kg BW) was given (i.p.) to reduce pain. Brain regions were identified according to the atlas[70]. After alignment of the brain in the stereotaxic surgery platform, 1–3 small holes were drilled in the skull at specific coordinates. The channelrhodopsin 2 (ChR2) and control virus (pAAV-EF1a-double floxed-hChR2(H134R)-EYFP-WPRE-HGHpA and pAAV-Ef1a-DIO EYFP, ~1 × 10$^{13}$ GC/ml, 333 nl, Addgene) was injected into LH unilaterally for optogenetic stimulation in LH or bilaterally for checking virus expression and optogenetic stimulation in raphe nuclei, with micropipettes pulled in house with a heating system. The coordinates from Bregma were: anterior-posterior, AP: −1.7 mm; medial-lateral, ML: ±0.9 mm; dorsal-ventral, DV: −5.0 mm. The GCaMP6 virus (AAV1.Syn.Flex.GCaMP6s.WPRE.SV40, ~4 × 10$^{12}$ GC/ml, 333 nl, Penn Vector Core) was injected to DR-ventral (AP: −4.6 mm, ML: 0 mm, DV: −3.0 mm) of 12–14-week-old Sert-Cre (control), Ox1R$^{\Delta SERT}$ and Ox2R$^{\Delta SERT}$ mice, to specifically express GCaMP6 in serotonergic neurons. After virus injection, optical fibers (fiber core = 200 µm, numerical AP = 0.48, flat tip; Doric lenses Inc.) were immediately implanted to stimulate LH (AP: −1.7 mm, ML: 0.9 mm, DV: −4.2 mm) or RPa (AP: −5.88 mm, ML: 0 mm, DV: −5.4 mm), and fixed to the skull with dental acrylic (Super-Bond C&B). Mice received a bolus of meloxicam (5 mg/kg BW, s.c.), and tramadol in drinking water for 3 d after surgeries, to relieve pain. BW was checked twice a day. Experiments started at least 4 weeks later than virus injection to allow virus expression. The optical fiber placement was histologically verified for each mouse at the end of the experiments.

**Retrograde tracing.** Pre- and post-surgery treatment of mice were the same as described above. After alignment of the brain in the stereotaxic surgery platform, two small holes were drilled in the skull for DR (AP: −4.6 mm, ML: 0 mm, DV: −2.8 to −3.2 mm) and RPa (AP: −5.88 mm, ML: 0 mm, DV: −5.8 to −6.1 mm) of 11-week-old BL/6 male mice. The red and green fluorescent RetroBeads™ (LumaFluor, Inc.) were injected with micropipettes into DR (500 nl) or RPa (120 nl) respectively. Five days later, mice were perfused and brains were further processed.

**Optogenetic stimulation in vivo.** Three weeks after virus injection, mice were transferred to the experimental cages for 1 week of habituation. At least 2 days before the experiments, a patch cord was connected to the optical fiber to allow acclimation. On the day of the experiment, a new patch cord replaced the old one. The laser was turned on 30 min before the injection of insulin or glucose, and the ITT and GTT were done as described above. The blue light (473 nm; 10 ms pulse, 20 Hz) was turned on for 10 s in every 45 s, and lasted until the end of ITT and GTT. Laser power was 20 mW to stimulate orexin neurons in LH and 10 mW to stimulate orexinergic fibers in RPa. The irradiance in the targeted region was above 2.5 mW/mm$^2$ (https://web.stanford.edu/group/dlab/cgi-bin/graph/chart.php), which was above the threshold to activate ChR2 (1 mW/mm$^2$) as reported by[71]. When measuring the response to insulin and glucose stimulation without laser illumination, mice were handled and experiments were done in the same way except for that the laser stayed off. There was 1 week of recovery between each experiment when using the same mice.

**Electrophysiology**

*Animals and brain slice preparation.* Experiments were performed on brain slices from 11 to 17-week old genetically marked (with tdTomato) Sert$^{tdTomato}$ (here referred to as control, Ctrl), Ox1R$^{\Delta SERT/tdTomato}$ and Ox2R$^{\Delta SERT/tdTomato}$ male mice. Animals were kept under standard laboratory conditions, with tap water and chow available ad libitum, on a 12 h light/dark cycle. The animals were lightly anesthetized with isoflurane (B506; AbbVie Deutschland GmbH and Co KG, Ludwigshafen, Germany) and decapitated. Coronal slices (280 µm) containing DR were cut with a vibration microtome (HM-650 V; Thermo Scientific, Walldorf, Germany) under cold (4 °C), carbogenated (95% O$_2$ and 5% CO$_2$), glycerol-based modified artificial cerebrospinal fluid (GaCSF)[72]. GaCSF contained (in mM): 244 Glycerol, 2.5 KCl, 2 MgCl$_2$, 2 CaCl$_2$, 1.2 NaH$_2$PO$_4$, 10 HEPES, 21 NaHCO$_3$, and 5 Glucose adjusted to pH 7.2 with NaOH. If not mentioned otherwise, the brain slices were continuously superfused with carbogenated aCSF at a flow rate of ~2.5 ml min$^{-1}$. aCSF contained (in mM): 125 NaCl, 2.5 KCl, 2 MgCl$_2$, 2 CaCl$_2$, 1.2 NaH$_2$PO$_4$, 21 NaHCO$_3$, 10 HEPES, and 5 Glucose adjusted to pH 7.2 with NaOH. To block GABAergic and glutamatergic synaptic input, the aCSF contained 10$^{-4}$ M PTX (picrotoxin, P1675; Sigma-Aldrich), 5 × 10$^{-6}$ M CGP (CGP-54626 hydrochloride, BN0597, Biotrend), 5 × 10$^{-5}$ M DL-AP5 (DL-2-amino-5-

phosphonopentanoic acid, BN0086, Biotrend), and 10$^{-5}$ M CNQX (6-cyano-7-nitroquinoxaline-2,3-dione, C127; Sigma-Aldrich).

*Patch-clamp recordings.* Perforated patch-clamp experiments were essentially performed as described previously[73,74]. Current-clamp recordings of tdTomato-expressing DR neurons were performed at ~32 °C. Neurons were visualized with a fixed stage upright microscope (BX51WI, Olympus, Hamburg, Germany) using ×40 and ×60 water-immersion objectives (LUMplan FL/N ×40, 0.8 numerical aperture, 2 mm working distance; LUMplan FL/N ×60, 1.0 numerical aperture, 2 mm working distance, Olympus) with infrared differential interference contrast optics[75] and fluorescence optics. tdTomato-expressing DR neurons were identified by their anatomical location and by their fluorescent label. Electrodes with tip resistances between 4 and 6 MΩ were fashioned from borosilicate glass (0.86 mm inner diameter; 1.5 mm outer diameter; GB150-8P; Science Products) with a vertical pipette puller (PP-830; Narishige, London, UK). All recordings were performed with an EPC10 patch-clamp amplifier (HEKA, Lambrecht, Germany) controlled by the program PatchMaster (version 2.32; HEKA) running under Windows. In parallel, data were recorded using a micro1410 data acquisition interface and Spike 2 (version 7) (both from CED, Cambridge, UK). Current clamp recordings were sampled at 25 kHz and low-pass filtered at 2 kHz with a four-pole Bessel filter.

Perforated patch experiments were conducted using protocols modified from before[76,77]. Recordings were performed with pipette solution containing (in mM): 140 K-gluconate, 10 KCl, 10 HEPES, 0.1 EGTA, 2 MgCl$_2$, and 1% biocytin (B4261, Sigma) adjusted to pH 7.2 with KOH. ATP and GTP were omitted from the intracellular solution to prevent uncontrolled permeabilization of the cell membrane[78]. The patch pipette was tip filled with internal solution and backfilled with internal solution, which contained the ionophore to achieve perforated patch recordings. Amphotericin B (A4888; Sigma) was dissolved in dimethyl sulfoxide to a concentration of 200 µg µl$^{-1}$ (DMSO; D8418, Sigma)[79] and added to the internal solution. The used DMSO concentration (0.1–0.3%) had no apparent effect on the investigated neurons. The final concentration of amphotericin B was ~120–160 µg µl$^{-1}$. Amphotericin solutions were prepared from undissolved weighted samples (stored at 4 °C protected from light) on every recoding day. During the perforation process, access resistance ($R_a$) was monitored continuously, and experiments started after $R_a$ had reached steady state (~15–20 min), and the action potential amplitude was stable.

Orexin A (ab120212, Abcam) or orexin B (O6262, Sigma) was added to the aCSF and bath-applied to the cells at one concentration or as an increasing concentrations series (1, 10, 100 nM) for 8–10 min each with a perfusion rate of ~2.5 ml min$^{-1}$.

**Ca$^{2+}$ imaging.** The experiments start at least 4 weeks after the GCaMP6 virus injection. Acute brain slices containing the DR were obtained as described for electrophysiology. Calcium dynamics were measured using the genetically encoded calcium indicator GCaMP6. The imaging setup consisted of a Zeiss AxioCam/MRm CCD camera with a 1388 × 1040 chip and a Polychromator V (Till Photonics, Gräfelfing, Germany) that was coupled via an optical fiber into the Zeiss AxioExaminer upright microscope (Objective W "Plan-Apochromat" 20x/1.0 DIC D = 0.17 M27 75 mm). The camera and polychromator were controlled by the software Zen pro, including the module 'Physiology' (2012 blue edition, Zeiss). The DRV neurons were identified according to their anatomical location and expression of the GCaMP6. Calcium signals in GCaMP6 expressing cells were monitored by images acquired at 470 nm excitation wavelengths with 80 ms exposure time at ~0.2 Hz. The emitted fluorescence was detected through a 500–550 nm bandpass filter (BP525/50), and data were acquired using 4 × 4 on-chip binning.

Orexin A and orexin B were applied for 10 min as described for electrophysiology. To analyze the orexin effect, we compared the fluorescence measured during 4 min intervals that were recorded immediately before and at the end of the peptide application. This protocol was followed by applying high K$^+$ concentration saline (40 mM KCl; osmolarity was adjusted by reducing the NaCl concentration accordingly) to elicit the maximal calcium response.

The image analysis was performed offline using Image J (version 1.53a) Igor Pro 6 and Prism 8 (GraphPad, California, USA). After the experiments, regions of interest (ROI) were defined, based on the high K$^+$ saline responses. The mean AU values of the ROIs were calculated in Image J. Time series analysis was performed with Igor Pro 6. To correct for bleaching artifacts, the baseline fluorescence (without orexin application) was fit. The extended fit was subtracted from the raw data. The orexin-induced Ca$^{2+}$ signals are given relative to the high K$^+$ response.

**Data analysis of electrophysiological and Ca$^{2+}$ imaging data.** Data analysis was performed with Spike2 (version 7; Cambridge Electronic Design Ltd., Cambridge, UK), Igor Pro 6 (Wavemetrics, Portland, OR, USA), and Graphpad Prism 8. If not stated otherwise, all calculated values are expressed as means ± SEM (standard error of the mean). The horizontal lines show the data's median. The whiskers were calculated according to the 'Tukey' method. For pairwise comparisons of dependent normal distributions, paired t-tests were used. For multiple comparisons, ANOVA with post hoc Tukey tests was performed. Tests were executed using GraphPad Prism 8. A significance level of 0.05 was accepted for all tests. Exact p-values are reported if $p > 0.05$. In the figures, n values are given in brackets.

The orexin effects were analyzed by comparing the action potential frequencies that were measured during 2 min intervals that were recorded before and at the end of the peptide applications. To analyze the orexin responsiveness, the neuron's firing rate averaged from 10 s intervals was taken as one data point. To determine the mean firing rate and standard deviation, 12 data points were averaged. On the single-cell level, a neuron was considered orexin-responsive if the change in firing induced by orexin was three times larger than the standard deviation (SD)[80,81].

**General statistical methods**. If not stated otherwise, all values are expressed as the mean ± SEM (standard error of the mean). Statistical analyses were conducted using GraphPad Prism 8. unless stated otherwise. Datasets with only two independent groups were analyzed for statistical significance using an unpaired two-tailed Student's $t$ test, and paired two-tailed $t$-test was used when data were matched. Datasets subjected to two independent factors were analyzed using two-way ANOVA followed by Sidak's post hoc test, if not stated otherwise. All $p$-values < 0.05 were considered significant (*,#$p < 0.05$, **,##$p < 0.01$, and ***,###$p < 0.001$, ****,####$p < 0.0001$).

**Reporting summary**. Further information on research design is available in the Nature Research Reporting Summary linked to this article.

## Data availability

The RNA-Seq data generated in this study have been deposited in the GEO database under accession code GSE168203. The source data underlying Figs. 1c–i, k–m, 2e–h, 3b, c, e, 4b–r, 5a–f, 6a–f, 7c–l, n–p, 8b, d, e, g, 9d, e, g and Supplementary Figs. 1j, n, o, p, r, 3c, 4a–h, 5a–f, 6a, b, 9a–d, f are provided as a Source Data file. Raw data that support the findings of this study are available from the corresponding author upon reasonable request. Publicly available datasets used in this study include GRCm38 dataset (https://www.ncbi.nlm.nih.gov/assembly/GCF_000001635.20) and "scRNA-seq_huang2019" (https://doi.org/10.7910/DVN/QB5CC8). There are no restrictions on data availability. Source data are provided with this paper.

## Code availability

The source code for analysing the RNA-Seq data presented in this study are deposited on GitHub with the repository bruening-lab/Sert-Ox1R (https://github.com/bruening-lab/Sert-Ox1R). There are no restrictions on code availability.

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

## Acknowledgements

We acknowledge Hella Brönneke for outstanding support; Jens Alber, Andreas Beyrau, Nadine Evers, Brigitte Hampel, Pia Scholl, Christiane Schäfer, Nadine Spenrath, Christian Heilinger, Helmut Wratil, and Kerstin Marohl for outstanding technical assistance; and Janine Altmüller and Marek Franitza from Cologne Center for Genomics for excellent technical assistance with RNA-seq, as well as Beatrix Martiny from CECAD Imaging Facility Cologne for excellent technical assistance with electron microscopy. We thank Dr. Henning Fenselau for critically reading the manuscript. This work was supported by a grant from the Deutsche Forschungsgemeinschaft (DFG) (BR 1492/7-1) to J.C.B., and we received funding by the DFG within the framework of the TRR 134 (A.C.H., J.C.B., P. Kloppenburg, F.T.W.) and within the Excellence Initiative by German Federal and State Governments (CECAD). X.X. received funding of CECAD family support. This work was funded (in part) by the Helmholtz Alliance ICEMED (Imaging and Curing Environmental Metabolic Diseases) through the Initiative and Networking Fund of the Helmholtz Association. Moreover, the research leading to these results has received funding from the European Union Seventh Framework Program (FP7/ 2007-2013) under grant agreement 266408. G.Y. gratefully acknowledges financial doctoral support from the DFG-233886668/GRK1960. B.B.L. is supported by the US National Institute of Health (NIH; R01 DK075632, R01 DK089044, R01 DK096010, R01 DK111401, P30 DK046200, and P30 DK057521). D.K. is supported by NIH (R01 NS107315, R01 DK108797, R21 HD098056, P30 DK046200). K.R. is supported by NIH (HL084207), the Department of Veterans Affairs (Merit grant BX004249), the University of Iowa Fraternal Order of Eagles Diabetes Research Center and the Iowa Neuroscience Institute.

## Author contributions

X.X., A.C.H., and J.C.B. conceived the project, designed the experiments, analyzed the data, and wrote the manuscript with input from the other authors. A.K. and F.T.W. designed the targeting strategy for conditional Ox1R flox mice and Ox2R flox mice and A.C.H. generated the mouse lines. B.B.L., T.E.S., and D.K. generated the Orexin-Cre mice. X.X. performed all the experiments apart from hyperinsulinemic-euglycemic clamp experiments (X.X., A.S., and A.C.H.), $Ca^{2+}$ imaging (G.Y. and P. Kloppenburg) and electrophysiological recordings (S.H., G.Y., and P. Kloppenburg). P. Klemm performed RNA sequencing data analysis. D.A.M. and K.R. helped with the design of BAT experiments, writing of the manuscript, and provided technical guidance. All authors discussed the data, commented on the manuscript before submission, and agreed with the final submitted manuscript.

## Funding

## Competing interests

The authors declare no competing interests.

**Additional information**

