## [Peer Review File · Nature Communications]

Reviewers' Comments:

Reviewer #1:

Remarks to the Author:

This study examined the role of orexin receptors in raphe nucleus with mouse models of specific deletion of orexin receptor 1 or 2 in serotonin neurons (SERT). Detailed analysis of both orexin receptor expression in serotonin neurons was performed, and a combination of mouse genetics, optogenetics, electrophysiology, fiber photometry, glucose clamping studies as well as extensive gene expression analysis was used to confirm and detail changes in the studied mouse models and metabolic phenotyping. Overall results show that orexin receptor 2 (OR2) expression in SERT neurons is required for normal glucose homeostasis through modulating liver glucose output. In addition, direction projection of orexin neurons to raphe pallidus was shown to improve glucose tolerance.

The strength of this manuscript is the detailed analysis of orexin receptor expression in SERT neurons and comprehensive analysis of gene expression in various peripheral tissues. The authors also went a great length in generating 2 new floxed mouse strains (OR1 floxed and OR2 floxed mice). However, overall this study appears to group a discrete sets of data with no apparent coherence and sometime conflicting implication, and therefore, it is rather difficult to draw a clear conclusion from these otherwise beautifully executed studies.

1) The overall physiological phenotype is rather subtle. Both KO mice exhibited a rather normal phenotype during chow diet and HFD diet and the only exception is that OR2 KOs exhibited an improved GTT after long time HFD. Given this subtle phenotype, it is perplexing that the authors identifying a rather impressive changes in gene expression in peripheral tissues. How do the authors explain with these changes, OR1 or OR2 KO failed to show changes in body weight in chow or HFD?

2) It is also perplexing how to reconcile the discrepancy between a rather impressive reduced GIR and glucose uptake in BAT and muscle (Fig 5 A-C), but no changes in GTTs or ITTs was observed in these mice? What is the significance of the glucose clamp studies?

3) How much of the effect of optogenetic stimulation of orexin fibers in raphe pallidus is related to orexin receptor effects or raphe pallidus? First of all, both OR1 and OR2 show minimal expression in this region. In addition, optogenetic stimulation of orexin fibers in this region may cause back-propagation, which may in turn cause orexin neuron activation. Thus, the impaired GTTs may be due to activation of orexin neurons in the lateral hypothalamus instead.

4) The authors analysed expression data from online database and found that the expression of orexin receptors is underestimated compared to their own RNAscope results. In this case, the colocalization between OR1 and OR2 might also be underestimated. Given the equally abundant OR1 and OR2 in DRD, there might be a significant degree of colocalization between the two, but needs further confirmation.

5) Given a relatively small percentage of OR1- or OR2-expressing SERT neurons, presumably a large of neurons recorded may not show responses to orexin A or B. How the authors differentiate the effect of orexin A or B on the recorded neurons is direct or indirect?

Reviewer #2:

Remarks to the Author:

Orexin (hypocretin) plays an important role in maintenance of wakefulness, but it is also involved in various physiological functions, including regulation of the autonomic nervous system. This study examined orexin's function of glucose homeostasis, focusing on its action on serotonergic neurons in the raphe nuclei. Authors first revealed that distributions of two subtypes of orexin receptors, OX1R and OX2R, in serotonergic subdivisions. They revealed that the dorsal raphe (DR) express both OX1R and OX2R, while the median raphe (MR) and raphe pallidus (RPa) almost exclusively expressed OX1R. Among DR populations OX1R is dominant in DRD, while OX2R is

dominant in DRV. They further show these receptor subtypes in serotonergic neurons play distinct, and opposite roles in glucose homeostasis by using genetic and optogenetic approaches. They used conditional KO mice that specifically lack OX1R or OX2R in serotonergic neurons to show these receptor subtypes play distinct roles. These mice did not show obvious phenotype in normal condition but showed different phenotype when induced HFD-induced obesity. Mice lacking OX1R in serotonergic neurons showed reduced insulin sensitivity in diet-induced obesity, through decreased glucose utilization in BAT and skeletal muscle. In contrast, deletion of OX2R in serotonergic neurons improved glucose tolerance and insulin sensitivity in obese mice due to decreased hepatic gluconeogenesis. These findings suggest that OX1R increases insulin sensitivity and glucose utilization through regulating BAT function, while OX2R rather decreases insulin sensitivity through regulating hepatic gluconeogenesis. In addition, they show optogenetic activation of cell bodies of orexin neurons in the LHA decreased glucose tolerance, while stimulation of orexin fibers in the RPa improved it. These data suggest distinct roles of orexin projections to serotonergic neurons in the different serotonergic subnuclei, and orexin receptor subtypes. This work is valuable for understanding role of orexin and its receptors in regulation of glucose homeostasis. However, several points should be addressed.

Major concerns

1. Authors used Sert-Cre BAC tg mice for manipulating serotonergic neurons. This line has been known to exhibit substantial level of ectopic expression which might affect result of the studies in many ways. Authors should examine how many tdTomato positive cells are actually serotonergic in Sert-tdTomato mice by examining expression of serotonergic markers such as TPH and Pet-1 or serotonin itself.
2. I guess orexin-Cre mice used in this study has not been described previously. If this is the first report using this line, authors should describe how this line was made, and characterize the phenotype, especially focusing on expression pattern of Cre activity.
3. Authors used OXR fl/fl mice without Cre expression as controls, but they should confirm fl/fl mice did not have any abnormalities by comparing them with wild type mice.
4. Authors show larger number of OX2R-only serotonergic neurons are VGLUT3-positive as compared with OX1R-only cells. VGLUT3-positive serotonergic neurons have been implicated in regulation of BAT function and thermoregulation. However, the present study rather showed involvement of OX1R in the RPa in BAT function. Please discuss.
5. Fos experiment on orexin neurons: Please state what time of the day you examined this. Authors only examined Fos expression in only one time point. But it is well known that activity of orexin neurons shows clear circadian fluctuation. They had better examine at several time points. Also it seems that Fos expression was increased not only by orexin neurons, but also by other non-orexin neurons. Please discuss this.
6. Optogenetic stimulation experiments showed soma stimulation of orexin neurons in the LHA decreased glucose tolerance, while fiber stimulation in the RPa rather increased it. This suggests orexin projections to the DRV, rather than projections to the RPa are dominant when orexin neurons in the LHA is activated. Please show the Fos expression patterns in the DRV, DRD and RPa after optogenetic manipulation in each condition. Also authors had better show Fos expression in orexin neurons in the LHA after optogenetic stimulation to reveal how many orexin neurons, and orexin neurons in which regions were activated by the manipulation.
7. This study suggests that the orexin->RPa pathway and orexin->DR pathway play distinct, and opposite roles in glucose homeostasis. Please discuss how these two distinct pathways are differentially regulated in physiological condition. Retrograde tracings from these regions to examine whether there are distinct orexin neuron populations that send innervations to these region might help.
8. Effects of orexin A on serotonergic neurons shown in Figure. 3 and Figure S2 are not impressive. Please show typical representative traces with longer time courses, showing baseline and wash-out phases in the same traces.

Minor concerns

- p.3, Ref. 3 should probably be changed to more recent one.
- p.5, Ref.5 did not mention about receptor subtype.
- p.8, ll.190-192: Radioligand binding assay. Show data or add reference.
- p.12, l.304, Name of the AAV vector, pAAV-hChR2-EYFP might cause misunderstanding. Because

this is a Cre-dependent vector, it should be noted as pAAV-DIO-hChR2-EYFP or something like that.

REVIEWER COMMENTS

Reviewer #1 (Remarks to the Author):

This study examined the role of orexin receptors in raphe nucleus with mouse models of specific deletion of orexin receptor 1 or 2 in serotonin neurons (SERT). Detailed analysis of both orexin receptor expression in serotonin neurons was performed, and a combination of mouse genetics, optogenetics, electrophysiology, fiber photometry, glucose clamping studies as well as extensive gene expression analysis was used to confirm and detail changes in the studied mouse models and metabolic phenotyping. Overall results show that orexin receptor 2 (OR2) expression in SERT neurons is required for normal glucose homeostasis through modulating liver glucose output. In addition, direction projection of orexin neurons to raphe pallidus was shown to improve glucose tolerance.

The strength of this manuscript is the detailed analysis of orexin receptor expression in SERT neurons and comprehensive analysis of gene expression in various peripheral tissues. The authors also went a great length in generating 2 new floxed mouse strains (OR1 floxed and OR2 floxed mice). However, overall this study appears to group a discrete sets of data with no apparent coherence and sometime conflicting implication, and therefore, it is rather difficult to draw a clear conclusion from these otherwise beautifully executed studies.

We thank the reviewer for carefully reading our manuscript. We feel that the comments/suggestions helped to improve our study as well as to identify a more coherent focus.

1) The overall physiological phenotype is rather subtle. Both KO mice exhibited a rather normal phenotype during chow diet and HFD diet and the only exception is that OR2 KOs exhibited an improved GTT after long time HFD. Given this subtle phenotype, it is perplexing that the authors identifying a rather impressive changes in gene expression in peripheral tissues. How do the authors explain with these changes, OR1 or OR2 KO failed to show changes in body weight in chow or HFD?

We absolutely agree with the reviewer that the overall metabolic changes are relatively mild. Nevertheless, we view this as an almost expected outcome, since the previously reported overall metabolic changes as observed upon manipulation of the entire orexin system are reproducible, yet not of extensive magnitude itself. Thus, in light of the fact, that we are investigating the effect of a specific output signal of the orexin system, i.e. to the serotonergic system, we find very clear tissue-specific effects of distinct OxR1- and OxR2-mediated pathways on a tissue level as well as during the well-defined clamp experiment, as outlined below. Moreover, we have included new analyses of GTTs after a 6hr fasting period (Figure 4 e, i, m, q), which is similar to the fasting condition during the clamp, revealing in conjunction more evidence for state-dependent differential regulation of also glucose tolerance in obesity depending on differential OxR1- and -2 signaling in serotonergic neurons in obesity. We have discussed this in the revised version of the manuscript. Collectively, we feel that defining these tissue-specific regulatory functions still significantly contributes to unraveling the detailed physiological function of the complex orexin system.

2) It is also perplexing how to reconcile the discrepancy between a rather impressive reduced GIR and glucose uptake in BAT and muscle (Fig 5 A-C), but no changes in GTTs or ITTs was observed in these mice? What is the significance of the glucose clamp studies?

As discussed above, the euglycemic hyperinsulinemic clamp is the gold standard for detailed assessment of whole body insulin sensitivity under very controlled conditions. The strength lays apparently in defining regulatory mechanisms, yet the overall regulation of glucose homeostasis as assessed during a GTT clearly depends on a whole array of organismal responses. Yet, the clamp allows to define the exact tissue-specific regulatory mechanisms and thus adding to our overall understanding of peripheral effects of very specific neurocircuits.

3) How much of the effect of optogenetic stimulation of orexin fibers in raphe pallidus is related to orexin receptor effects or raphe pallidus? First of all, both OR1 and OR2 show minimal expression in this region. In addition, optogenetic stimulation of orexin fibers in this region may cause back-propagation, which may in turn cause orexin neuron activation. Thus, the impaired GTTs may be due to activation of orexin neurons in the lateral hypothalamus instead.

In raphe pallidus, we have observed that Ox1R was expressed in more than 60% of serotonergic neurons while no Ox2R was expressed (Figure 1 and 2), and most of the Ox1R signal was in serotonergic neurons. Figure 9 also shows that orexinergic fibers very well colocalize with serotonergic neurons. Though back-propagation possibly exists, we found GTT **impaired** by optical stimulation of orexin neurons in LH but **improved** by optical stimulation of orexinergic fibers in raphe pallidus, which made it unlikely the effect was due to back-propagation. Therefore, we think the effect of optical stimulation of orexinergic fibers in raphe pallidus is mostly related to Ox1R in serotonergic neurons. We apologize, if we haven't sufficiently clarified this issue in the first submission.

4) The authors analysed expression data from online database and found that the expression of orexin receptors is underestimated compared to their own RNAscope results. In this case, the colocalization between OR1 and OR2 might also be underestimated. Given the equally abundant OR1 and OR2 in DRD, there might be a significant degree of colocalization between the two, but needs further confirmation.

We completely agree with the reviewer and accordingly have counted the co-localization of Ox1R and Ox2R in serotonergic neurons with RNAscope and added the data in Figure 1 (j, k). We found 39% of co-localization instead of 2% shown by scRNAseq. Of note, we think the scRNAseq analysis is still meaningful, for investigation of the Ox1R- or Ox2R- dominant serotonergic neurons in DR. We further counted the VGLUT3 expression in Ox1R-only and Ox2R-only serotonergic neurons (Figure 1 j, l, m), and confirmed that more Ox2R-only neurons (88%) expressed VGLUT3 than Ox1R-only neurons (13%), though the numbers are not completely the same as in scRNAseq (55% and 14%, respectively). Very likely this is accounted for by the fact that lowly abundant expression of GPCRs such as the OxRs likely escapes detection in single cell sequencing experiments given their current limitation in sequencing depths.

5) Given a relatively small percentage of OR1- or OR2-expressing SERT neurons, presumably a large of neurons recorded may not show responses to orexin A or B. How the authors differentiate the effect of orexin A or B on the recorded neurons is direct or indirect?

As the reviewer suggested in comment 4), there is underestimation of orexin receptor positive SERT neurons in scRNAseq analysis. With RNAscope experiments, we found 79% of Ox1R- and/or Ox2R-positive SERT neurons in DR (Figure 1 j, k), which is consistent with the number we showed in

Figure 1 (a, c, e). Here we also analyzed the number of responders in E.phys. and it matches to our findings using RNAscope (Figure 3b,c). As we mentioned in the methods and results parts, the electrophysiological recordings were performed in the presence of GABA and glutamate receptor blockers, i.e. upon isolation from synaptic input, further supporting that the effect of orexin A and B is direct.

Reviewer #2 (Remarks to the Author):

Orexin (hypocretin) plays an important role in maintenance of wakefulness, but it is also involved in various physiological functions, including regulation of the autonomic nervous system. This study examined orexin's function of glucose homeostasis, focusing on its action on serotonergic neurons in the raphe nuclei. Authors first revealed that distributions of two subtypes of orexin receptors, OX1R and OX2R, in serotonergic subdivisions. They revealed that the dorsal raphe (DR) express both OX1R and OX2R, while the median raphe (MR) and raphe pallidus (RPa) almost exclusively expressed OX1R. Among DR populations OX1R is dominant in DRD, while OX2R is dominant in DRV. They further show these receptor subtypes in serotonergic neurons play distinct, and opposite roles in glucose homeostasis by using genetic and optogenetic approaches. They used conditional KO mice that specifically lack OX1R or OX2R in serotonergic neurons to show these receptor subtypes play distinct roles. These mice did not show obvious phenotype in normal condition but showed different phenotype when induced HFD-induced obesity. Mice lacking OX1R in serotonergic neurons showed reduced insulin sensitivity in diet-induced obesity, through decreased glucose utilization in BAT and skeletal muscle. In contrast, deletion of OX2R in serotonergic neurons improved glucose tolerance and insulin sensitivity in obese mice due to decreased hepatic gluconeogenesis. These findings suggest that OX1R increases insulin sensitivity and glucose utilization through regulating BAT function, while OX2R rather decreases insulin sensitivity through regulating hepatic gluconeogenesis. In addition, they show optogenetic activation of cell bodies of orexin neurons in the LHA decreased glucose tolerance, while stimulation of orexin fibers in the RPa improved it. These data suggest distinct roles of orexin projections to serotonergic neurons in the different serotonergic subnuclei, and orexin receptor subtypes. This work is valuable for understanding role of orexin and its receptors in regulation of glucose homeostasis. However, several points should be addressed.

We thank the reviewer for his/her positive comments on our initial submission.

Major concerns

1. Authors used Sert-Cre BAC tg mice for manipulating serotonergic neurons. This line has been known to exhibit substantial level of ectopic expression which might affect result of the studies in many ways. Authors should examine how many tdTomato positive cells are actually serotonergic in Sert-tdTomato mice by examining expression of serotonergic markers such as TPH and Pet-1 or serotonin itself.

Concerning the reviewer's comments, we have systemically analyzed the Cre expression patterns and added the results in Figure S1.

Specifically, we have analyzed Cre expression using a Sert-tdTomato mice. Beyond the raphe nuclei, tdTomato was also expressed in deep layers of cingulate cortex (Cg1 and Cg2), and ventral posterolateral (VPL) and ventral posteromedial (VPM) thalamic nuclei (Figure S1 a-h). However, there was no detectable orexin receptor expression in tdTomato positive cells in these areas except for moderate expression levels of Ox2R in cingulate cortex, while abundant Ox1R and Ox2R were expressed in tdTomato positive cells in DR (a positive control), as revealed by RNAscope in situ

hybridization (Figure S1 i-l). We further quantified the co-expression of Ox2R and tdTomato in cingulate cortex and found that most of the Ox2R signal was not in tdTomato positive cells and there were 10.74% (5 cells in 45 cells) of tdTomato positive cells expressing Ox2R (Figure S1 i, j). Therefore, we do not expect ectopic expression of Cre in these areas would significantly affect our findings in this study.

In addition, we have stained TPH2, Pet1 and tdTomato in raphe nuclei of Sert-tdTomato mice using RNAscope. Almost all tdTomato positive neurons are TPH2 and/or Pet1 positive in DR, MR and RPa, and we could hardly see tdTomato single labeled cells (Figure S1 m, n, p, q). More than 95% of Tph2 and/or Pet1 positive neurons are tdTomato positive (Figure S1 m, o, q, r). This indicates that Cre expression is specific and efficient in serotonergic neurons in raphe nuclei.

2. I guess orexin-Cre mice used in this study has not been described previously. If this is the first report using this line, authors should describe how this line was made, and characterize the phenotype, especially focusing on expression pattern of Cre activity.

The reviewer is absolutely right, and we apologize for the lack of a more detailed description of this newly generated mouse in the initial submission. We have added a description of how the mouse line has been generated in the revised material & methods section. We have examined the Cre expression pattern using orexin^{Chr2-tdTomato} mice, which were obtained by crossing Orexin-Cre mice and Chr2-tdTomato fl/fl mice. The data is now added in Figure S7. tdTomato was specifically expressed in orexin A-positive cells in LH, though there was a small cluster of tdTomato single labeled cells below the lateral ventricle and around the 3rd ventricle and several scattered cells in other brain regions. Therefore, we have decided to inject a Cre-dependent Chr2-EYFP virus in LH of Orexin-Cre mice in our optogenetic experiments. We confirmed the specificity and efficiency of virus expression in Figure 8 (a, b).

3. Authors used OXR fl/fl mice without Cre expression as controls, but they should confirm fl/fl mice did not have any abnormalities by comparing them with wild type mice.

We agree with the reviewer that fl/fl mice can show a different phenotype when compared to wild type mice, though we did not observe obvious disabilities or abnormalities of our fl/fl mouse lines in past years. Trying to avoid this possible effect, we have always used littermates for control (SERT-Cre +/-; OXR fl/fl) and knockout (SERT-Cre +/-; OXR fl/fl) groups.

It would be best to breed for fl/fl and BL/6 wild type mice in the same facility and perform the metabolic analysis in same cohorts. Due to time and practical limitations, we have decided to insist on our experimental design for the control in this study. We hope the reviewer agrees that, in our present manuscript, using littermates is reasonable and enough to investigate the effect of specific deletion of OXR in serotonergic neurons.

4. Authors show larger number of OX2R-only serotonergic neurons are VGLUT3-positive as compared with OX1R-only cells. VGLUT3-positive serotonergic neurons have been implicated in regulation of BAT function and thermoregulation. However, the present study rather showed involvement of OX1R in the RPa in BAT function. Please discuss.

In the initial manuscript, the analysis of scRNAseq data of DR serotonergic neurons revealed more Ox2R-only serotonergic neurons are VGLUT3 positive compared with Ox1R-only cells. In our revised version of the manuscript, we could again confirm this finding with RNAscope and we added the results in Figure 1 (j, l, m). Previous reports on the regulation of BAT function by VGLUT3-positive serotonergic neurons focused on the raphe pallidus but not DR. These findings are consistent with our results that orexinergic fibers projecting to serotonergic neurons in RPa are important for BAT function.

5.Fos experiment on orexin neurons: Please state what time of the day you examined this. Authors only examined Fos expression in only one time point. But it is well known that activity of orexin neurons shows clear circadian fluctuation. They had better examine at several time points. Also it seems that Fos expression was increased not only by orexin neurons, but also by other non-orexin neurons. Please discuss this.

Fos expression in orexin neurons has been measured under 6-h fasting conditions, because the metabolic effects revealed in our study (by GTT and euglycemic-hyperinsulinemic clamp) were all detected under fasting conditions but not randomly-fed conditions (in ITT). Beyond circadian fluctuation, orexin neurons are also activated by fasting. Therefore, we have decided to keep the fasting condition when checking Fos expression in orexin neurons.

Furthermore, we agree, that in LH, there are also other neurons beyond orexin neurons, which respond to HFD feeding. For example, excitatory synapses to MCH neurons increased upon HFD feeding (Linehan V., et al., 2020, see below). However, we would like to focus on the orexin system in this study. After comprehensive considerations, we did not quantify other Fos positive cells in LH or discuss this in the main text.

Ref.

Linehan, V., Fang, L.Z., Parsons, M.P., and Hirasawa, M. (2020). High-fat diet induces time-dependent synaptic plasticity of the lateral hypothalamus. *Mol Metab* 36, 100977.

6.Optogenetic stimulation experiments showed soma stimulation of orexin neurons in the LHA decreased glucose tolerance, while fiber stimulation in the RPa rather increased it. This suggests orexin projections to the DRV, rather than projections to the RPa are dominant when orexin neurons in the LHA is activated. Please show the Fos expression patterns in the DRV, DRD and RPa after optogenetic manipulation in each condition. Also authors had better show Fos expression in orexin neurons in the LHA after optogenetic stimulation to reveal how many orexin neurons, and orexin neurons in which regions were activated by the manipulation.

We have analyzed Fos expression in LH and raphe nucleus for each condition in our optogenetic experiments. Fos-positive orexinergic or serotonergic neurons significantly increase after optical stimulation of orexin neurons in LH or orexinergic fibers in RPa, respectively. We have added the data in Figure 8 (f, g) and 9 (f, g). The activated orexin neurons are located from medial to lateral and we did not observe obvious region specificity. This is in line with our optical fiber implantation, which was 1 mm above the orexin neurons, and we aimed to spread the light in a broader field to activate most of the orexin neurons.

We could not detect significant changes of Fos activation in serotonergic neurons in DRD, DRV or RPa after soma stimulation in LH, and we consider two possible reasons for that: 1) inhibition from

indirect pathways and/or 2) technical limitations of Fos staining. We have included the figure below for reviewer's inspection. In general, only scattered Fos signal is seen in serotonergic neurons. With the difficulty to highlight the true signal by background subtraction, it is hardly to detect signal changes unless the change is dramatic (which happened in the direct optical stimulation in RPa). We had similar experiences with Fos staining of serotonergic neurons in raphe nucleus for other experimental purpose before, and we think it may not be the optimal way to investigate the activity of serotonergic neurons. For long run, it will be better to use E.Phys. or GCaMP.

c-Fos expression in raphe nuclei after optical stimulation of orexin neurons in LH. (a, c) Representative images of RNAscope *in situ* hybridization of c-Fos and serotonin transporter (SERT) in dorsal raphe nucleus-dorsal (DRD) and -ventral (DRV) and raphe pallidus (RPa). (b, d) Quantification of percentages of c-Fos positive neurons in serotonergic neurons after laser illumination. n = 4. Scale bar: 200 μ m (a) or 100 μ m (c). Data are represented as means \pm SEM.

7. This study suggests that the orexin->RPa pathway and orexin->DR pathway play distinct, and opposite roles in glucose homeostasis. Please discuss how these two distinct pathways are differentially regulated in physiological condition. Retrograde tracings from these regions to examine whether there are distinct orexin neuron populations that send innervations to these region might help.

We thank the reviewer for his/her very important suggestions. We think retrograde tracings are a very good strategy to analyze the underlying pathway mechanisms. Accordingly, we have injected red retrobeads in DR and green retrobeads in RPa region, and found distinct and intermingled green and red fluorescent cells in LH. We added the data in Figure 8 h-j. This indicates intermingled and distinct LH cells project to DR or RPa. The neurochemical heterogeneity of orexin neurons (Mickelsen L.E., et al., 2017) may help to understand their functional heterogeneity. Unfortunately, we could not identify the cell specificity due to technical reasons. In long run, cell specific retrograde tracing using rabies virus can be used. Likewise, it has been reported that rostral-VTA- and caudal-VTA-projecting

orexin neurons in LH respond differentially to reward, and no topographic segregation was reported (Richardson K.A., et al., 2012).

Mickelsen, L.E., Kolling, F.W., IV, Chimileski, B.R., Fujita, A., Norris, C., Chen, K., Nelson, C.E., and Jackson, A.C. (2017). Neurochemical Heterogeneity Among Lateral Hypothalamic Hypocretin/Orexin and Melanin-Concentrating Hormone Neurons Identified Through Single-Cell Gene Expression Analysis. *eNeuro* 4, ENEURO.0013–17.2017.

Richardson, K.A., and Aston-Jones, G. (2012). Lateral Hypothalamic Orexin/Hypocretin Neurons That Project to Ventral Tegmental Area Are Differentially Activated with Morphine Preference. *Journal of Neuroscience* 32, 3809–3817.

8. Effects of orexin A on serotonergic neurons shown in Figure. 3 and Figure S2 are not impressive. Please show typical representative traces with longer time courses, showing baseline and wash-out phases in the same traces.

We have included representative traces in Supplementary Figure 3 a, b.

Minor concerns

p.3, Ref. 3 should probably be changed to more recent one.

We thank the reviewer for his/her suggestion. We have changed the reference accordingly (p.3, Ref. 3).

p.5, Ref.5 did not mention about receptor subtype.

We would like to apologize for this mistake. We have changed the text accordingly and thank the reviewer for having brought this to our attention (p.3, Ref. 5).

p.8, ll.190-192: Radioligand binding assay. Show data or add reference.

We have added the reference in the main text (line 2 on p.9).

p.12, l.304, Name of the AAV vector, pAAV-hChR2-EYFP might cause misunderstanding. Because this is a Cre-dependent vector, it should be noted as pAAV-DIO-hChR2-EYFP or something like that.

We thank the reviewer for bringing this to our attention. We have changed the text accordingly (line 15-17 on p. 13).

Once again, we thank the reviewers for their very constructive and fair input to our study, and we feel that by addressing them through numerous additional experiments and changes to the text, the manuscript has significantly improved.

Reviewers' Comments:

Reviewer #1:

Remarks to the Author:

I thank for the effort from the authors in addressing my concerns. However, I feel the following 2 major concerns have not been adequately addressed.

1) The overall significance of this study : while the authors provided a lot of data in describing the animal models, expression and phenotyping, the overall conclusion appears to be rather limited. The major phenotype appears to be limited to mild GTT in OXR2 KOs. The rest phenotypes on body weight (both chow and HFD) and others appear to be roughly normal. Although the CLAMPS results show some differences, given the normal phenotype (especially for OXR1), the physiological significance of the CLAMPS data is not immediately clear. Also gene expression and morphological differences observed in the peripheral tissues appear to be inconsistent with the overall normal phenotypes on body weight and glucose homeostasis (except GTTs in OXR2) in both chow and HFD.

2) Issues on back-propagation: the authors' explanation is not satisfactory. It is possible that selective activation of the subset of Orexin neurons that project to raphe pallidus produce an opposite effect to the overall activation of LH Orexin neurons. Back-propagation to this subset of orexin neurons may activate additional downstream sites other than raphe pallidus that produces that the observed GTT phenotype. Addressing this issue will also help to address the apparent opposite phenotypes in glucose homeostasis mediated by OXR1 and OXR2 in raphe.

Reviewer #2:

Remarks to the Author:

Authors did a good work to address this reviewer's concerns, and I am almost totally satisfied with their revision, which includes several additional data. Manuscript improved significantly. Especially, authors carefully examined expression pattern of tdTomato-positive Cells in Sert-tdTomato mice and showed results in Figure S1. Although they found many ectopic expressions, the authors revealed that expression levels of orexin receptors in these ectopically tdTomato expressing neurons were minimal. They also beautifully showed tdTomato-positive neurons in the raphe nuclei were mostly serotonergic. These data are important and have strengthened the authors' original claims. The authors also did retrograde tracing to reveal that orexin neuronal populations that send projections to DR and RPa are distinct.

Authors added description about how orexin-Cre mouse line was generated in the method section, with showing expression pattern of Cre in this line in Fig. S7, which is also important.

REVIEWER COMMENTS

Reviewer #1 (Remarks to the Author):

I thank for the effort from the authors in addressing my concerns. However, I feel the following 2 major concerns have not been adequately addressed.

We have addressed the remaining major concerns on the physiological significance of our study with additional textual changes in the discussion and inclusion of a paragraph on limitations of the study, and the technical concerns on the back-propagation issue with additional experiments.

1) The overall significance of this study: while the authors provided a lot of data in describing the animal models, expression and phenotyping, the overall conclusion appears to be rather limited. The major phenotype appears to be limited to mild GTT in OXR2 KOs. The rest phenotypes on body weight (both chow and HFD) and others appear to be roughly normal. Although the CLAMPS results show some differences, given the normal phenotype (especially for OXR1), the physiological significance of the CLAMPS data is not immediately clear. Also gene expression and morphological differences observed in the peripheral tissues appear to be inconsistent with the overall normal phenotypes on body weight and glucose homeostasis (except GTTs in OXR2) in both chow and HFD.

We agree with the reviewer, that we did not detect any significant changes in peripheral tissues, glucose homeostasis or body weight of conditional OxR knockout mice under NCD-fed conditions. All phenotypes were only detected in conditional OxR knockout mice under HFD-fed conditions. We think this indicates that orexin signaling in serotonergic neurons could be more crucial for maintaining energy homeostasis under obese conditions, compared to lean conditions. Consistent with this hypothesis, we found orexin neurons were more activated in HFD-fed mice compared to CD-fed mice under fasting conditions. We have discussed this in the revised manuscript (highlighted on p15-16).

Under HFD-fed conditions, the major phenotype of Ox1R^{ASERT} and Ox2R^{ASERT} mice appears to be limited to GTT and CLAMPS, both of which were performed under fasting conditions. Gene expression, protein expression and/or morphological differences were observed in peripheral tissues (liver, BAT and SM) at the end of the Clamp-experiment under very controlled conditions, and in BAT of random-fed obese Ox1R^{ASERT} mice sacrificed at the age of 21 weeks. The overall metabolic effect of orexin signaling in serotonergic neurons seems to be state-dependent. We cannot systemically investigate BAT in obese Ox1R^{ASERT} mice on a time-scale due to practical reasons. The clear changes in BAT could be late-onset and the time of HFD exposure may not have been sufficient to translate these effects into significant changes in BW or body composition, while compensation of other peripheral organs is also possible. We have discussed this in detail in the revised manuscript (highlighted on p17-18).

Clearly, further well-designed studies are needed to investigate the underlying mechanisms of the above findings. Though we could not address all the mechanisms in detail with the present manuscript, we hope

the reviewer agrees that our findings still significantly contribute to unravel the physiological functions of the complex orexin and serotonergic systems.

2) Issues on back-propagation: the authors' explanation is not satisfactory. It is possible that selective activation of the subset of Orexin neurons that project to raphe pallidus produce an opposite effect to the overall activation of LH Orexin neurons. Back-propagation to this subset of orexin neurons may activate additional downstream sites other than raphe pallidus that produces that the observed GTT phenotype. Addressing this issue will also help to address the apparent opposite phenotypes in glucose homeostasis mediated by OXR1 and OXR2 in raphe.

We agree with the reviewer and think that these are important concerns. To investigate this, we measured c-Fos in LH orexin neurons of mice after optical stimulation of orexinergic fibers in RPa. We couldn't detect changes in LH orexin c-Fos expression between Orexin^{Chr2-EYFP} and Orexin^{EYFP} mice after optical stimulation in RPa. We have added a new Supplemental Figure (Supplementary Fig. 9 e, f) and an updated Supplemental file in the revised manuscript. Based on these new results, we hope the reviewer agrees that back-propagation of a subset of orexin neurons is most likely not producing the observed phenotype.

Reviewer #2 (Remarks to the Author):

Authors did a good work to address this reviewer's concerns, and I am almost totally satisfied with their revision, which includes several additional data. Manuscript improved significantly. Especially, authors carefully examined expression pattern of tdTomato-positive Cells in Sert-tdTomato mice and showed results in Figure S1. Although they found many ectopic expressions, the authors revealed that expression levels of orexin receptors in these ectopically tdTomato expressing neurons were minimal. They also beautifully showed tdTomato-positive neurons in the raphe nuclei were mostly serotonergic. These data are important and have strengthened the authors' original claims. The authors also did retrograde tracing to reveal that orexin neuronal populations that send projections to DR and RPa are distinct.

Authors added description about how orexin-Cre mouse line was generated in the method section, with showing expression pattern of Cre in this line in Fig. S7, which is also important.

We thank the reviewer for carefully reading our manuscript and his/her positive comments on our study.

Once again, we thank the editors and reviewers for their very thoughtful comments on our study, and we feel that by addressing the remaining concerns with additional experiments and textual changes, the manuscript has further significantly improved.

Very much hoping for your positive response, I remain,

Sincerely,

Christine Hausen, PhD

Reviewers' Comments:

Reviewer #1:

Remarks to the Author:

The authors have adequately addressed my concerns.

Reviewer #1 (Remarks to the Author):

The authors have adequately addressed my concerns.

We thank the reviewer for carefully reading our manuscript and his/her positive comments on our study.